behaviour, cognition

rational monitoring, metacognitive monitoring, great apes, children, decision-making, information-seeking

**Author for correspondence:**
Cathal O'Madagain
e-mail: cathal.omadagain@um6p.ma

# Great apes and human children rationally monitor their decisions

Cathal O'Madagain[1,2], Katharina A. Helming[1,3], Marco F. H. Schmidt[1,4], Eli Shupe[5], Josep Call[1,6] and Michael Tomasello[1,7]

[1]Department of Developmental and Comparative Psychology, Max Planck Institute for Evolutionary Anthropology, Leipzig, Germany
[2]School of Collective Intelligence, Université Mohammed VI Polytechnique, Ben Guérir, Morocco
[3]Department of Psychology, University of Warwick, Coventry, UK
[4]Department of Psychology, University of Konstanz, Germany
[5]Department of Philosophy and the Humanities, University of Texas at Arlington, Arlington, TX, USA
[6]School of Psychology and Neuroscience, University of St Andrews, St Andrews, UK
[7]Department of Psychology, Duke University, Durham, NC, USA

 CO, 0000-0002-4086-524X; MFHS, 0000-0001-7651-1681; MT, 0000-0002-1649-088X

Several species can detect when they are uncertain about what decision to make—revealed by opting out of the choice, or by seeking more information before deciding. However, we do not know whether any nonhuman animals recognize when they need more information to make a decision because new evidence contradicts an already-formed belief. Here, we explore this ability in great apes and human children. First, we show that after great apes saw new evidence contradicting their belief about which of two rewards was greater, they stopped to recheck the evidence for their belief before deciding. This indicates the ability to keep track of the reasons for their decisions, or 'rational monitoring' of the decision-making process. Children did the same at 5 years of age, but not at 3 years. In a second study, participants formed a belief about a reward's location, but then a social partner contradicted them, by picking the opposite location. This time even 3-year-old children rechecked the evidence, while apes ignored the disagreement. While apes were sensitive only to the conflict in physical evidence, the youngest children were more sensitive to peer disagreement than conflicting physical evidence.

## 1. Introduction

The ability to tell when one cannot make a reliable decision has been found in several species [1–11]. Sometimes called 'metacognitive monitoring', it is often tested in 'opt-out' experiments, where participants decline difficult decisions in favour of easier ones. It is also tested in information-seeking experiments. When great apes are presented with two containers, where only one holds a reward, they will stop and look inside the containers to be sure where the reward is before choosing one. This effort to get exactly the information they need shows that they 'know what they do not know' ([8–11], although see [12,13]).

While these studies reveal the ability to detect when one has no knowledge, they do not reveal an ability to think about what one already believes, which is a different kind of metacognition. Thinking about what one believes is sometimes elicited when one encounters new evidence that calls a prior belief into question. Suppose you believe it is sunny out and plan to go to the beach. Now the dog comes into the house soaking wet. This may prompt you to look out the window to recheck your reason for believing it is sunny before deciding to leave. In such a scenario, we are aware of what we believe, and we check the grounds or reason for that belief against what the new evidence indicates. Following terminology used by philosophers [14], this can be called 'rational' or 'reason-based' monitoring of the decision-making process.

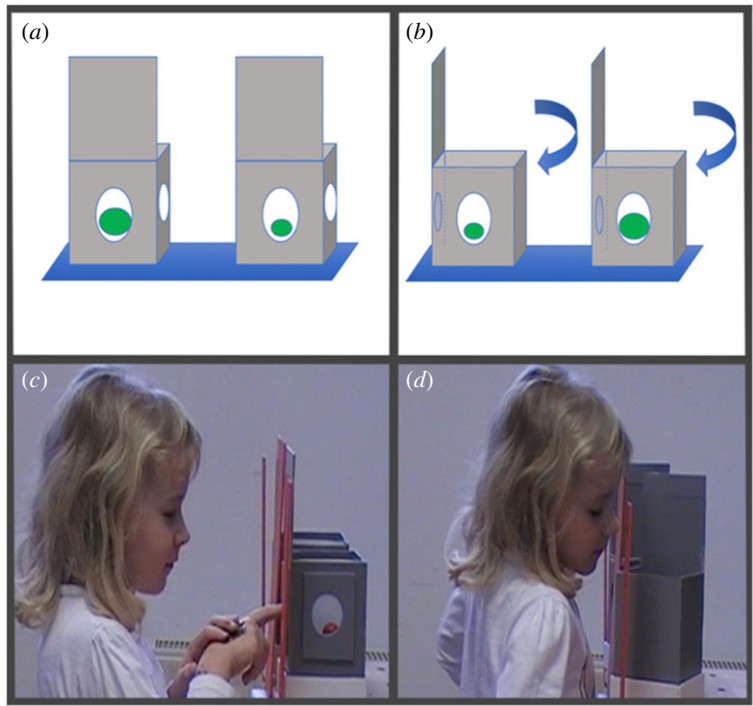

**Figure 1.** The conflicting physical evidence apparatus. On the 'first view' one reward looks bigger than the other (*a*). On the 'second view' (in the 'conflicting' condition), the opposite reward appears to be bigger (*b*); (*c*) a participant makes a choice given the 'first view' and looks for more information before choosing on the 'second view' (*d*). (Online version in colour.)

Here, we devised a new methodology to explore this ability. In a first study, we tasked participants with evaluating two contradictory pieces of physical evidence. Apes and children were presented with an initial piece of evidence leading them to form a belief about which of two rewards was greater, which they indicated with a choice. Then, before being rewarded, new equally good evidence appeared, indicating the opposite reward was greater. Now participants could stop and look for more information before making a final decision, or they could choose again without looking. We expected that participants that recognized the contradiction between their prior belief and the new evidence would seek more information before choosing.

For humans, such contradictions arise more commonly via disagreements with a social partner—one person believes it is sunny but another says it is raining. Our second study explored this kind of contradiction. After participants formed a belief about the location of a reward, an onlooking social partner gestured towards an opposite location as the site of the reward. Here, treating the conflict as grounds for uncertainty requires understanding not only that our own belief might be false, but also that our partner may be mistaken [15–18]. Because human problem-solving is adapted for social contexts [19,20], we expected the youngest children to take this 'social contradiction' as equal or stronger grounds for uncertainty than the contradictory physical evidence of the first study. We expected this task would be less compelling for apes, however, whose cognition has evolved primarily for individual problem-solving [21].

## 2. Study 1: evaluating conflicting physical evidence

In our first study, were apes (*n* = 18), 3-year-old (*n* = 64) and 5-year-old children (*n* = 66). We chose these age groups because

children's understanding of belief develops substantially between 3 and 5 years [15–18], so that by including both we might identify a developmental change. Participants were presented with two boxes, with windows cut into the sides (figure 1). Each contained a reward—one bigger than the other. After making an initial choice, the boxes were rotated 90° to reveal a second 'view' of their contents. In 'consistent' trials, the rewards appeared the same on both views. In 'conflicting' trials, owing to the use of magnifying/minimizing lenses, the relative size of the rewards appeared to reverse on the second view (the larger now appearing smaller). Participants could now seek extra information before making a final choice, by peeking inside the boxes from the top (figure 1*d*; electronic supplementary material, movie S1). Or, they could make a final choice without peeking (electronic supplementary material, movie S2).

To ensure participants knew they could check for more information, we ran 'warm-up' trials, where participants could not see the rewards through windows and could only make a decision by peeking inside. Participants who did not peek in these trials were excluded.

### (a) Analysis

We used generalized linear mixed models (GLMMs) with binomial error structure to test the data (electronic supplementary material, S6–S9; models and data available at https://osf.io/ey5f9/). A full-null comparison was significant ($\chi^2 = 22.291$, $p = 0.001$), where the null model lacked the terms 'population' (3 years, 5 years and apes) and 'condition' (consistent and conflicting), and their interaction. Model comparison revealed no significant interaction between population and condition ($\chi^2 = 6.86$, $p = 0.334$). Therefore, we removed the interaction term and tested for an effect of 'condition'. This revealed a significant overall effect of condition ($\chi^2 = 16.095$, $p < 0.0001$). It appears,

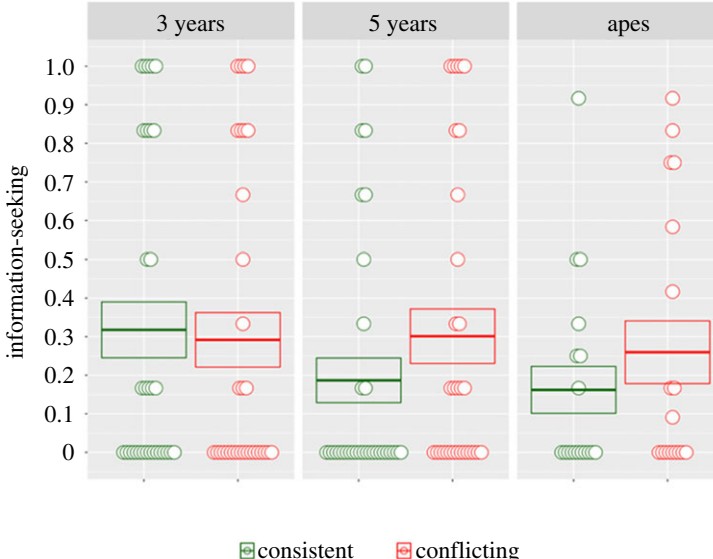

**Figure 2.** Results of the conflicting physical evidence task. Apes and 5-year-old children sought additional information more when faced with conflicting than consistent evidence, but 3-year olds did not. Discs represent individual averages across trials; the number of discs at any point on the *y*-axis represents the distribution of responses. Boxes represent means and standard errors. (Online version in colour.)

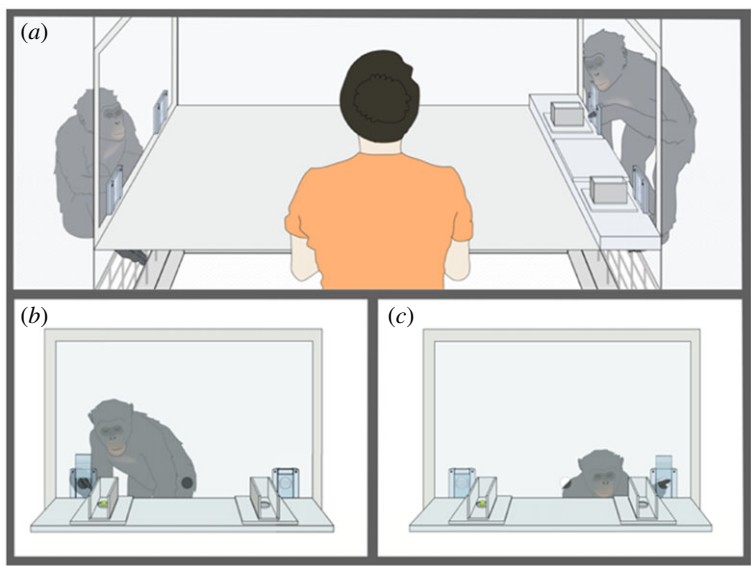

**Figure 3.** The conflicting opinions' apparatus. The experimenter deposits the reward in one of two boxes with both participants watching. The non-target chooses first (*a*), then the target is presented with the choice. In the 'conflicting' condition, after the experimenter deposits the reward, it is hidden in a hole cut in the floor of the box, while the other box is loaded with an identical reward before the trial begins. When the non-target participant sees inside the boxes, she therefore sees the reward in the opposite location from where the target saw it deposited and chooses that location. The target can peek before choosing (*b*), or choose without peeking (*c*). (Online version in colour.)

however, that the effect of condition is driven by the apes and 5-year olds, rather than the 3-year olds (who peeked slightly more in the consistent condition) (figure 2).

When presented with new evidence that conflicts with a prior belief based on equally good evidence, great apes sought additional information before making a decision (see also the electronic supplementary material, figure S1). This ability appears to have emerged by the age of 5 years in human children, but not clearly by the age of 3 (a further analysis, reported below after study 2, supports this interpretation).

## 3. Study 2: evaluating conflicting opinions

In a second study, participants were faced with conflicting evidence that came from the opinion of a social partner, rather than new physical evidence. Participating in this study were apes (*n* = 17), 3-year-old (*n* = 72) and 5-year-old children (*n* = 42). Participants faced each other and between them were two boxes on a slider that could be moved back and forth (figure 3). At the start of each trial, the experimenter deposited a reward into one of the boxes, in view of both participants. The experimenter then rotated the boxes so that an open side became visible to the non-target participant and pushed the boxes towards the non-target (figure 3*a*). In 'consistent' trials, the non-target picks what the target expects, but in 'conflicting' trials (owing to a surreptitious manipulation, figure 3), the non-target sees the reward in the opposite location from where it was initially deposited, and picks that location. The non-target's choice now conflicts with what the target expected, generating 'peer disagreement' on the location of the reward.

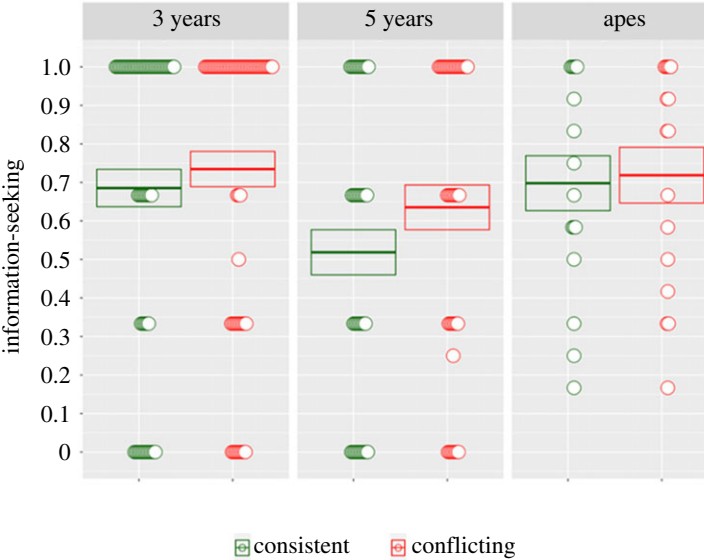

**Figure 4.** Results of the conflicting opinions' task. Children at both 3 and 5 years looked for extra information more when a peer's opinion conflicted with their own; apes do not appear to be contributing to the effect (notice the standard error bars are overlapping the mean in the apes but not in either group of children). Discs represent individual averages across trials. Boxes represent means and standard errors. (Online version in colour.)

The boxes are now moved to the target who can take a peek inside before choosing (figure 3b) or can choose without peeking (figure 3c).

## (a) Analysis

We used a GLMM to measure information-seeking by population and condition. A comparison of a full model with a null model lacking the terms population, condition and their interaction was significant ($\chi^2 = 14.867$, $p = 0.01$), allowing us to reject the null hypothesis. A comparison of the full model to a reduced model lacking the interaction term showed no significant improvement ($\chi^2 = 2.3665$, $p = 0.306$). We therefore removed the interaction term and compared models with and without the term 'condition', finding a significant overall effect ($\chi^2 = 7.659$, $p = 0.005$): subjects peeked more in response to a contradictory opinion than one that was consistent with their prior belief (figure 4).

The results of our second study indicate that children as young as 3 years take peer disagreement as a reason to doubt their prior beliefs. There was no interaction between populations, but this time the effect appears to be driven primarily by the children rather than the apes (see also the electronic supplementary material, figure S2). Our next analysis further supports this interpretation.

Overall we had predicted that great apes would be more sensitive to conflicting physical rather than social information, while young children would be more sensitive to conflicting social information. To test this, we pooled all data and tested for a three-way interaction of population, condition and study—to see if different groups were more sensitive to the distinction between conditions from one study to the other. This was significant ($\chi^2 = 11.408$, $p = 0.003$). To isolate the source of the three-way interaction, we tested for an interaction between study and condition in each population separately. We found an interaction between condition and study in the apes ($\chi^2 = 4.312$, $p = 0.037$): apes distinguished the conditions more in the physical study than in the social study. We also found an interaction between study and condition in the 3-year olds ($\chi^2 = 11.596$, $p < 0.001$): 3-year olds distinguished the conditions

more in the social study than in the physical study. There was no interaction in the 5-year olds, but rather a main effect of condition: the 5-year olds peeked significantly more when faced with conflicting evidence in both studies ($\chi^2 = 13.746$, $p < 0.001$) (see the electronic supplementary material, table S1 for an overview). These results support our original hypothesis: apes were more sensitive to conflicting physical evidence than social evidence, while the youngest children were more sensitive to conflicting social than physical evidence.

One challenge raised for studies like this one is that participants may be considering just one piece of evidence at a time, rather than considering old and new evidence together (Leahy & Carey [22]). Here this would mean participants were thinking only of the new evidence when they sought more information, rather than comparing their prior belief to the new evidence—and that would not be metacognition. To investigate this, we checked where participants looked first when they peeked. If participants only considered the new evidence, then they should be just as likely to first check the location indicated by the new evidence in both conditions—as the new evidence is identical in both conditions. On the other hand, if participants are considering their prior belief and the new evidence at once, then when the two conflict (conflicting), they should be more likely to first check in the opposite location to that indicated by the new evidence, than when the two indicate the same location (consistent). We found that in the 'conflicting' trials participants were indeed more likely to peek first in the opposite location to that indicated by the new evidence than in the 'consistent' trials ($z = 7.93$; $p < 0.0001$) (electronic supplementary material, figure S3). This is hard to explain if participants were not thinking of their prior belief and the new evidence at once.

Another concern is that in the social study, the apes may not be attending to their partner's choices. This would explain why they were slower to recheck evidence in light of conflicting opinions than conflicting physical evidence. To rule this out, we conducted an 'ignorance-knowledge' post-test. Here, participants received no prior information about the location of the reward, but could see that their partner could see where it was. Now participants significantly followed their partners'

choices (5 year olds mean = 0.91, $p < 0.0001$; 3 year olds mean = 0.88, $p < 0.0001$; apes mean = 0.63, $p < 0.001$) (electronic supplementary material, figure S4). This rules out the possibility that the apes were not paying attention to their partners in the social study. Apes knew what choices their partners were making, but peer disagreement was not enough to get them to doubt their prior belief.

## 4. General discussion

Previous studies have shown that several species will look for more information when they do not have enough to answer a question [1–11], and that young children will update their beliefs to match new evidence [23–27]. However, our studies show for the first time, to our knowledge, that apes and young children seek more information when old and new evidence conflict, but are equally strong. Rather than simply updating their earlier belief to match the new evidence, they double-check the evidence—checking first in the location indicated by their prior belief. The intuitive explanation for this behaviour is that participants knew what they believed, and sought to compare the reason for their prior belief with what the new evidence told them, recognizing that either could be wrong. They were, in other words, examining the reasons for their belief or 'rationally monitoring' the decision-making process. Apes were more sensitive to conflicting physical evidence rather than peer disagreement, while young children were more sensitive to peer disagreement.

A concern that similar studies have been unable to rule out is that participants may be considering just one piece of evidence at a time rather than reconsidering their prior belief in light of the new information (Leahy & Carey [22]). Here, this would mean that participants were only thinking of the new evidence when they sought more information. As seen above, however, in the 'conflicting' trials, participants were significantly more likely to check the opposite location to that indicated by the new evidence than in the 'consistent' trials, which is best explained by supposing they were still thinking of their prior belief. Another possible concern is that when faced with conflicting information, involuntary hesitation might become a cue that participants use to learn that whenever they experience it, they should look for more information [13]. Or, we might worry that information-seeking could be generated by a decision taking too long, triggering a sort of 'reset' to return to foraging [28]. On these views, participants would not 'know what they do not know', but engage in automatic information-seeking triggered by a cue. Arguably if this were true we should expect that they would search randomly for food when their information-seeking began. However, this was not the case—participants looked for exactly the information they needed to resolve the conflict (checking the containers). This 'targeted information-seeking' [3] is best explained by supposing that participants know what they do

not know, seeking the information they need to specifically address this question.

Recognizing that reliable decisions cannot be made on the basis of contradictory reasons has long been considered to be the cornerstone of rationality. Aristotle argued that only creatures capable of recognizing contradictory reasons as poor grounds for a decision could count as rational animals [29]. The philosophers List and Pettit define a rational agent as one that possesses the ability to prevent herself from acting on contradictory beliefs [14]; and the rejection of contradictory beliefs is considered to be a key to the economic model of rational decision-making [30]. That apes display this ability suggests that the distinction of being a *rational animal* does not easily demarcate humans from all other species. One of the hallmarks of rationality—the ability to weigh contradictory reasons against one another—is shared by humans and great apes. On the other hand, these studies taken together suggest that the major distinction between human and great ape decision-making is in its sociality. Younger children are more likely to look for more information given peer disagreement than conflicting physical evidence, while apes are more likely to double-check given a conflict in physical evidence rather than peer disagreement. This fits with the view that human rationality is adapted for social purposes, including solving peer disputes, argumentation and knowledge transmission [19–21,31,32]. Our findings show that humans are not just *good* at problem-solving in social contexts, but *better* at solving problems socially than individually—and that the distinguishing mark of human cognition is its sociality.

Ethics. These studies were non-invasive and adhered to the legal requirements of the country in which they were conducted. They were approved by the Max Planck Institute for Evolutionary Anthropology Ethics Committee (members of the committee are Prof. M. Tomasello, head of the child laboratory Katharina Haberl and research assistant Jana Jurkat). The procedure of the study was covered by the committee's approval. Consent was obtained from the parents of the children who participated in this study.

Data accessibility. Data, statistical models, analyses and sample photographs of material used are available online from the Open Science Foundation: https://osf.io/ey5f9/.

Authors' contributions. C.O.: conceptualization, data curation, formal analysis, investigation, methodology, project administration, writing—original draft and writing—review and editing; K.A.H.: conceptualization, methodology and writing—review and editing; M.F.H.S.: conceptualization, methodology and writing—review and editing; E.S.: investigation and methodology; J.C.: conceptualization, methodology and writing—review and editing; M.T.: conceptualization, methodology and writing—review and editing.

All authors gave final approval for publication and agreed to be held accountable for the work performed therein.

Competing interests. We declare we have no competing interests.

Funding. We received no external funding for this study.

Acknowledgements. Thanks to Suzanne Mauritz for her coordination of data collection in the children's study, Johannes Grossmann for his work on the ape study, Colleen Stephens who advised on statistical analysis and Laura Daerr for illustrations.

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
