## [Peer Review File · Proceedings of the Royal Society B: Biological Sciences]

Review History

RSPB-2021-1076.R0 (Original submission)

Review form: Reviewer 1

Recommendation

Major revision is needed (please make suggestions in comments)

Scientific importance: Is the manuscript an original and important contribution to its field?

Good

General interest: Is the paper of sufficient general interest?

Good

Quality of the paper: Is the overall quality of the paper suitable?

Acceptable

Is the length of the paper justified?

No

Should the paper be seen by a specialist statistical reviewer?

No

Do you have any concerns about statistical analyses in this paper? If so, please specify them explicitly in your report.

Yes

It is a condition of publication that authors make their supporting data, code and materials available - either as supplementary material or hosted in an external repository. Please rate, if applicable, the supporting data on the following criteria.

Is it accessible?

No

Is it clear?

N/A

Is it adequate?

N/A

Do you have any ethical concerns with this paper?

No

Comments to the Author

The authors present an interesting study suggesting that there are differences in children and apes' ability to monitor their confidence and seek information when a peer acts in a direction that is in conflict with their own belief. I think study2 is interesting, but that there are a few alternative interpretations of the results at the moment, and further analysis could potentially allow the authors to rule them out. The manuscript also needs some work in order to better situate the results with respect to past research.

Major

« we do not know whether any nonhuman animals recognize when they cannot make a reliable decision because they have several conflicting pieces of information. » ; “what has not been investigated is whether young children or animals can detect their own uncertainty because they have conflicting information »

We do know a lot about this, e.g., see Hampton 2009; Kornell 2007; Kepecs et al., 2012; Rosati et Santos 2016... In many animal studies on metacognition, one could argue that there is ambiguity in the sensory evidence that forces the animal to weigh competing options... The authors argue that past research has looked at cases where no information is present, but there are actually a lot of perceptual decision studies where animals need to weigh 2 response options as a function of ambiguous sensory evidence (e.g., see reviews by Smith et al., 2012; Hampton 2009; Kepecs et al., 2012; Proust 2019...). Similarly, in children studies on metacognitive monitoring often rely on presenting ambiguous sensory evidence (e.g., see Lyons and Ghetti 2013; Geurten & Basin 2018; Beran et al., 2012; Baer and Odic 2021; Vo et al., 2014...). Arguably, when faced with a binary choice an agent will always weight competing options and conflicting evidence? The fact that a lot of the relevant literature on metacognition in children and animals is not covered seems problematic... It seems that the main novelty here concerns the findings about meta-reasoning (e.g., see Ackerman & Thompson 2017) in the face of social contradiction (i.e., study 2). Some reworking of the introduction/abstract seems required to clarify what is novel here (e.g., shifting the focus on the novel findings about social contradiction, and discuss at greater length the inter-species differences and what they may mean)?

It would also be important to discuss points raised e.g., by Hampton 2009; Perner and Dienes 2013; Leahy and Carey 2020..., that in this kind of information seeking task it is difficult to know whether agents engage in metacognitive monitoring, rather than trying options sequentially, and/or engaging in information seeking automatically when no response comes to mind. For

instance, Hampton (2009, also see Perner & Dienes 2013) has suggested that information seeking in this type of task could be driven by response competition, as a form of default option triggered when the decision-making fails after a certain period of time (i.e., a sort of “time is up” automatic response). Can the authors provide more data analysis (for instance an analysis of response times for peeking versus choosing, e.g., see Goupil et al., 2016) that could allow them to rule out this response competition interpretation? Another important point to address is the possibility that apes/children do not really weight conflicting information here, but instead estimate how probable 1 option is at a time (e.g., see Leahy and Carey 2020). Here nothing comes to my mind in terms of additional data analysis that could rule out this interpretation, but there may be some aspects in the data that speak against this possibility? If not some interpretations will have to be readjusted in function (especially the conclusions of the paper).

« engaging in curiosity-driven information seeking can be explained simply by being more interested in objects that behave in novel ways »: how is this supposed to work without metacognitive monitoring...? If curiosity is a desire for information (i.e., excludes other types of foraging), and that children engage in curiosity-driven information seeking when they are presented with novel or conflicting information because they realize that this does not conform to their prior knowledge, or that they cannot make sense of it, how can this be explained without them registering at some level that this activity could lead to a cognitive gain because they lack information? (i.e., without hypothesizing that there is a basic form of metacognitive monitoring involved); this point seems to contradict the rest of the argument upon which the paper is based... making a distinction between metarepresentations / explicit metacognition versus more basic forms of metacognition that do not necessarily require full-fledged, conceptual metarepresentations could be one way to go here (e.g., see Proust 2007; 2012; Shea et al., 2014; Goupil et al., 2019...). Another way to go is to not enter this debate about what curiosity requires at all.

The results are interesting in relation to the suggestion that explicit metacognition has a social function (Shea et al., 2014) and it could be nice to discuss the inter-species differences/similarities in this respect?

Study 1:

“This ability has emerged by age five in human children, but not at three”: it would be good to show the performances of the children and apes in choosing; where 3yo worse/slower? this could explain the difference in peeking in this age group... (i.e., a difference driven by task performances and/or motivation rather than metacognition) ; as mentioned above previous research suggests that 3yo can monitor their ability to discriminate between 2 options, but only when they are competent with the task at hand. A possible interpretation here is that 3yo were not competent enough in the discrimination task, and/or that they were not motivated enough to peek. Figure S4 suggests that in fact 3yo peek a lot, which suggests low confidence, where in general younger children have a bias of overconfidence when task performances are controlled for... So perhaps this suggests that 3yo are not so competent in this task (while ruling out the possibility that they are less motivated to peek overall). Other studies (some mentioned above) also suggest that young children may be better at monitoring their own knowledge and refrain from responding when they lack information when they are engaged in a social interaction... More discussion on this seems required.

Study 2: the interaction is not significant between apes, 3 and 5yo so it seems difficult to interpret the inter-species difference... was there an interaction if testing an effect of population considering only apes vs. children? Relatedly, would it not make sense to use population (children, apes) as a factor, and then test age effects by using only children data throughout the paper?

Minor

Study 1: I am guessing the format of the journal prevents the authors from describing the paradigm at greater length, but still a lot of useful information seems to be missing from the

description of the set-up on p.4 (e.g., how were participants shown that they could peek when seeing the second display but not before? how many trials? what is “order”? how was peeking coded...); quite a lot of the information that is now in the SM seems required to allow readers to assess the findings.

I am not sure I understand the model comparisons, was there no significant interaction between condition and population when comparing nested models like these:

peeking ~ condition + population

peeking ~ condition * population

Please clarify, this is ambiguous at the moment... It may be easier to present the results of sequentially nested models (e.g., comparing the null model with a model including only condition, then adding population, then the interaction term or something like this).

p.5: “due to a surreptitious manipulation” what is this manipulation? it would be useful to describe this (e.g., in the Figure caption)

p.7: the paper would be easier to read if the comparison between study 1 and 2 was moved in the results section (now the first sentence of the discussion seems a bit unjustified at first because we haven’t seen the results at that point)

Review form: Reviewer 2 (Stephen Ferrigno)

Recommendation

Major revision is needed (please make suggestions in comments)

Scientific importance: Is the manuscript an original and important contribution to its field?

Excellent

General interest: Is the paper of sufficient general interest?

Excellent

Quality of the paper: Is the overall quality of the paper suitable?

Good

Is the length of the paper justified?

Yes

Should the paper be seen by a specialist statistical reviewer?

Yes

Do you have any concerns about statistical analyses in this paper? If so, please specify them explicitly in your report.

Yes

It is a condition of publication that authors make their supporting data, code and materials available - either as supplementary material or hosted in an external repository. Please rate, if applicable, the supporting data on the following criteria.

Is it accessible?

Yes

Is it clear?

Yes

Is it adequate?

Yes

Do you have any ethical concerns with this paper?

No

Comments to the Author

Review: Great Apes and Human Children Rationally Monitor Their Decisions

Comments to the Authors:

The authors present a novel and very interesting way of testing apes' & young children's ability to seek information after receiving conflicting information. The design well thought out and is a good test of the question at hand. This manuscript would have broad appeal in comparative cognition, developmental psychology, and cognitive psychology. The authors conclude that apes seek information after receiving conflicting information directly, but not when the conflicting information is given via a social partner. In contrast, three-year-olds looked more when the competing information is from a social partner. The manuscript is well written and clear. The one large issue I have is that the statistics used do not directly test the conclusions made in the paper (see below). However, given the graphs in the supplemental, I think the data is there and will likely lead to the same conclusions. With some major revisions, the paper would be a great fit for Proc. B.

1. The term "Rational Decision Making" seems a bit of a stretch given the evidence. There seems like two reasonable possibilities for how the information from different sources is combined.
 - 1) "comparing those competing piece of evidence or 'reasons' for making the decision against one another", the reason the authors presume.
 - 2) An alternative is that the animals & children put some type of weight on the information they receive immediately and then just compare these weights. There would be no need to think about where each of the evidence comes from or to do any type of "reason-based" monitoring once the evidence has been given a weight.

While this seems like a small point, it is at the heart of the argument in the paper would benefit from further clarification whether the authors think possibility #2 is still rational (reason-based) or not.

2. Study 1 analysis: The method of using a GLMM works well for one of the conclusions drawn from the analyses (that collapsed across all groups there was an effect of condition overall effect of condition across all groups). However, it is not fit for some of the conclusions made (e.g. which group is driving the overall effect of condition). The finding that there was no interaction between population and condition exemplifies this issue. To then go on and say that the effect is driven by any subset of these groups (e.g. apes/5yos) is unfounded given the statistics presented.
 - a. There are two reasonable ways to approach this: The first is to use contrast coding within the larger model to look at specific differences between the groups, the other is to run three separate, simpler models on each of the groups.
 - b. In the full model was "condition" significant before removing all of the included variables?
 - c. Given that the data is clearly not normally distributed, graphing the means for each group/condition does not seem like the most informative graph. A violin plot would highlight the wide variety of responses that were given. OR the type of plots used in figure S1/S2.
 - i. Furthermore, on this point a statement about how many animals and children are driving these effects seems necessary. It is clearly not the case that all of the animals look when they get conflicting information (at least 9/18 monkeys, 15/66 5yos, & 18/64 3yos never looked)
 1. How many children/monkeys looked more often in the conflicting case?
 2. How many actually looked more in the consistent case?

d. Given the above questions about the data, the conclusions that “Great apes can detect their own uncertainty... when the reasons for their decision are in conflict.” and “This ability has emerged by age five in human children”, seems like a stretch given that at least 15 5yos never looked and half of the apes never looked. Reporting how many subjects in each group looked more often (compared to how many looked less often) when there was conflicting info is necessary for these conclusions.

3. Study 2 analysis: My comments about the study 1 analysis all apply to this analysis as well. The only conclusion that can be made given the statistics used is that when all subjects (regardless of group) are collapsed, there is an effect of condition

a. Here the conclusion that 3yos are now successful is unfounded given the statistics used (as is the finding that apes failed).

b. All of the groups seem to have a bias to not look in experiment 1 (regardless of condition), and a bias to look in experiment 2.

i. What is the reason for this?

ii. Why is the effect size between the difference tasks (social/not social) much larger than any of the effect seen in condition (consistent/not consistent)?

4. General Discussion

a. The authors address the differences between the specific groups, but do so in a roundabout way that does not completely fix the issues raised in the analysis sections.

i. The statistic used shows there was an interaction effect between experiment & condition in apes only shows that they peeked more in the physical condition compared to the social condition. A better way to do this would be to compare each of these to chance, i.e. is there a significant effect of condition in the physical experiment for apes? Is there a difference between conditions in the social experiment?

1. The above also applies for the conclusion about 3yos.

ii. Given that these findings are the base of the entire study, more direct analyses are needed.

b. The explanation for why 3-year-olds are better in the social condition is unclear. It is not the case that experiment 2 was a 100% social task. It was a combination of a social and a physical task. Subjects did not just need to pay attention to social cues, but to the direct physical evidence as well (in which they failed to do in experiment 1). How/why do children pass on the version that integrates social and physical evidence when they fail on physical evidence alone. The explanation that they are better in social-contexts seems incomplete.

Minor comments:

- Paragraph 2/line 1: There has also been work showing that monkeys can tell when they have some, but not enough information (see Brady & Hampton, 2021), this is beyond know they have no information.
- Were there any differences between ape species?
- SM/methods & materials: Were the subjects that were excluded counted in the subject count in SM:Participants and the main text? I assume not, but a clarification would be good.
- What box did the subjects choose when they did not look in the inconsistent trials?
- Did they choose at random? Or did they choose the most recent information, e.g. the box that showed the item to be larger after the turn?
- Table S1. Seems uninformative as is. Instead of broad conclusions, including the direct statistics in these boxes would be much better. E.g. each of the boxes represent the results of a model using “looking” as the dependent variable and condition as the predictor. That way you have 6 models each summarized in a box rather than a conclusion without any stats to back it up.

References:

Brady, R. J., & Hampton, R. R. (2021). Rhesus monkeys (*Macaca mulatta*) monitor evolving decisions to control adaptive information seeking. *Animal Cognition*, 1-9.

-Stephen Ferrigno

Review form: Reviewer 3

Recommendation

Major revision is needed (please make suggestions in comments)

Scientific importance: Is the manuscript an original and important contribution to its field?

Acceptable

General interest: Is the paper of sufficient general interest?

Good

Quality of the paper: Is the overall quality of the paper suitable?

Acceptable

Is the length of the paper justified?

Yes

Should the paper be seen by a specialist statistical reviewer?

Yes

Do you have any concerns about statistical analyses in this paper? If so, please specify them explicitly in your report.

Yes

It is a condition of publication that authors make their supporting data, code and materials available - either as supplementary material or hosted in an external repository. Please rate, if applicable, the supporting data on the following criteria.

Is it accessible?

Yes

Is it clear?

Yes

Is it adequate?

Yes

Do you have any ethical concerns with this paper?

No

Comments to the Author

The paper presents some clever manipulations that reveal some interesting aspects of human and nonhuman information-seeking behavior. But, it is seriously marred by an unnerving and unjustified predilection toward over-interpretation and using mentalistic notions when simpler behavioral interpretations would suffice. Many terms and expressions are set in quotation marks leaving the reader to wonder if these are scientifically suitable notions or just window dressing to attract greater public attention.

The paper would definitely profit by the authors taking a more circumspect approach and more seriously considering the manipulations and findings from a behavioral perspective. After all, we

only have what can be observed and confirmed--the rest is sheer speculation.

Furthermore, the paper is not well written. There are a few too many grammatical and punctuation errors as well as a serious mislabeling in the caption of Fig. S1. More careful preparation of a revision would certainly be in order.

Finally, there is something of an unholy blending of within-subject and between-subject manipulations finding their way into the same statistical analyses. There is also concern that the two studies might not properly be included in a single statistical analysis. And, there are very few individuals from some ape species in the studies and their data are also quite disparate; can they all legitimately be lumped into a single 'ape' group?

Decision letter (RSPB-2021-1076.R0)

15-Jun-2021

Dear Professor O'Madagain:

I am writing to inform you that your manuscript RSPB-2021-1076 entitled "Great Apes and Human Children Rationally Monitor Their Decisions" has, in its current form, been rejected for publication in Proceedings B.

This action has been taken on the advice of referees, who have recommended that substantial revisions are necessary. With this in mind we would be happy to consider a resubmission, provided the comments of the referees are fully addressed. However please note that this is not a provisional acceptance.

Sincerely,

Dr Robert Barton

Associate Editor

Comments to Author:

This is a study of metacognition in apes and children. Understanding of metacognitive information seeking has emerged as an important cognitive skill with repercussions for many aspects of mature human behaviors, and this set of studies develop a clever way to address its developmental and phylogenetic roots, and further introduces a novel distinction about the source of conflicting information (social versus physical) that may then lead to information-seeking. The reviewers (and I) agree that this is an interesting topic and novel experimental paradigm, but also have helpful comments regarding the paper that should be addressed in revision.

One important point concerns the statistics and interpretation of the results. For example, R1 and R2 point out several comments about how the current statistical reporting does not always support the conclusions drawn from the studies, with the most salient being that the main outcome of Study 2 is that there is a main effect of condition (and no improvement by including the interaction with population) but the interpretation is written as though there was a significant interaction. R1 and R3 also comment about the particularities of how these populations (younger and older children versus apes) are grouped in analyses, and the paper would benefit from more clearly explaining the logic here. The reviewers have several other helpful comments about statistical approaches and best practices for reporting that should be addressed. Better clarity on the use of GLMMs will likely also address comments from R3 about within- versus between-subjects manipulations

Another important point is to provide a more robust theoretical framework for the study that better incorporate vies from cognitive science exploring the particular mechanisms that may underpin information seeking. R1 provides some helpful ideas, references and other possible interpretations that could be helpful to this end to provide a more nuanced interpretation of the actual cognitive processes underlying responses in the task. R2 also points out similar comments related to the use of the term 'rational decision making.' I believe a careful treatment of these issues will also help address R3's concerns about mentalistic interpretation of the results.

Reviewer(s)' Comments to Author:

Referee: 1

Comments to the Author(s)

The authors present an interesting study suggesting that there are differences in children and apes' ability to monitor their confidence and seek information when a peer acts in a direction that is in conflict with their own belief. I think study2 is interesting, but that there are a few alternative interpretations of the results at the moment, and further analysis could potentially allow the authors to rule them out. The manuscript also needs some work in order to better situate the results with respect to past research.

Major

« we do not know whether any nonhuman animals recognize when they cannot make a reliable decision because they have several conflicting pieces of information. » ; "what has not been investigated is whether young children or animals can detect their own uncertainty because they have conflicting information »

We do know a lot about this, e.g., see Hampton 2009; Kornell 2007; Kepecs et al., 2012; Rosati et Santos 2016... In many animal studies on metacognition, one could argue that there is ambiguity in the sensory evidence that forces the animal to weigh competing options... The authors argue that past research has looked at cases where no information is present, but there are actually a lot of perceptual decision studies where animals need to weigh 2 response options as a function of ambiguous sensory evidence (e.g., see reviews by Smith et al., 2012; Hampton 2009; Kepecs et al., 2012; Proust 2019...). Similarly, in children studies on metacognitive monitoring often rely on

presenting ambiguous sensory evidence (e.g., see Lyons and Ghetti 2013; Geurten & Basin 2018; Beran et al., 2012; Baer and Odic 2021; Vo et al., 2014...). Arguably, when faced with a binary choice an agent will always weight competing options and conflicting evidence? The fact that a lot of the relevant literature on metacognition in children and animals is not covered seems problematic... It seems that the main novelty here concerns the findings about meta-reasoning (e.g., see Ackerman & Thompson 2017) in the face of social contradiction (i.e., study 2). Some reworking of the introduction/abstract seems required to clarify what is novel here (e.g., shifting the focus on the novel findings about social contradiction, and discuss at greater length the inter-species differences and what they may mean)?

It would also be important to discuss points raised e.g., by Hampton 2009; Perner and Dienes 2013; Leahy and Carey 2020..., that in this kind of information seeking task it is difficult to know whether agents engage in metacognitive monitoring, rather than trying options sequentially, and/or engaging in information seeking automatically when no response comes to mind. For instance, Hampton (2009, also see Perner & Dienes 2013) has suggested that information seeking in this type of task could be driven by response competition, as a form of default option triggered when the decision-making fails after a certain period of time (i.e., a sort of "time is up" automatic response). Can the authors provide more data analysis (for instance an analysis of response times for peeking versus choosing, e.g., see Goupil et al., 2016) that could allow them to rule out this response competition interpretation? Another important point to address is the possibility that apes/children do not really weight conflicting information here, but instead estimate how probable 1 option is at a time (e.g., see Leahy and Carey 2020). Here nothing comes to my mind in terms of additional data analysis that could rule out this interpretation, but there may be some aspects in the data that speak against this possibility? If not some interpretations will have to be readjusted in function (especially the conclusions of the paper).

« engaging in curiosity-driven information seeking can be explained simply by being more interested in objects that behave in novel ways »: how is this supposed to work without metacognitive monitoring...? If curiosity is a desire for information (i.e., excludes other types of foraging), and that children engage in curiosity-driven information seeking when they are presented with novel or conflicting information because they realize that this does not conform to their prior knowledge, or that they cannot make sense of it, how can this be explained without them registering at some level that this activity could lead to a cognitive gain because they lack information? (i.e., without hypothesizing that there is a basic form of metacognitive monitoring involved); this point seems to contradict the rest of the argument upon which the paper is based... making a distinction between metarepresentations / explicit metacognition versus more basic forms of metacognition that do not necessarily require full-fledged, conceptual metarepresentations could be one way to go here (e.g., see Proust 2007; 2012; Shea et al., 2014; Goupil et al., 2019...). Another way to go is to not enter this debate about what curiosity requires at all.

The results are interesting in relation to the suggestion that explicit metacognition has a social function (Shea et al., 2014) and it could be nice to discuss the inter-species differences/similarities in this respect?

Study 1:

"This ability has emerged by age five in human children, but not at three": it would be good to show the performances of the children and apes in choosing; where 3yo worse/slower? this could explain the difference in peeking in this age group... (i.e., a difference driven by task performances and/or motivation rather than metacognition) ; as mentioned above previous research suggests that 3yo can monitor their ability to discriminate between 2 options, but only when they are competent with the task at hand. A possible interpretation here is that 3yo were not competent enough in the discrimination task, and/or that they were not motivated enough to peek. Figure S4 suggests that in fact 3yo peek a lot, which suggests low confidence, where in general younger children have a bias of overconfidence when task performances are controlled for... So perhaps this suggests that 3yo are not so competent in this task (while ruling out the

possibility that they are less motivated to peek overall). Other studies (some mentioned above) also suggest that young children may be better at monitoring their own knowledge and refrain from responding when they lack information when they are engaged in a social interaction... More discussion on this seems required.

Study 2: the interaction is not significant between apes, 3 and 5yo so it seems difficult to interpret the inter-species difference... was there an interaction if testing an effect of population considering only apes vs. children? Relatedly, would it not make sense to use population (children, apes) as a factor, and then test age effects by using only children data throughout the paper?

Minor

Study 1: I am guessing the format of the journal prevents the authors from describing the paradigm at greater length, but still a lot of useful information seems to be missing from the description of the set-up on p.4 (e.g., how were participants shown that they could peek when seeing the second display but not before? how many trials? what is "order"? how was peeking coded...); quite a lot of the information that is now in the SM seems required to allow readers to assess the findings.

I am not sure I understand the model comparisons, was there no significant interaction between condition and population when comparing nested models like these:

peeking ~ condition + population

peeking ~ condition * population

Please clarify, this is ambiguous at the moment... It may be easier to present the results of sequentially nested models (e.g., comparing the null model with a model including only condition, then adding population, then the interaction term or something like this).

p.5: "due to a surreptitious manipulation" what is this manipulation? it would be useful to describe this (e.g., in the Figure caption)

p.7: the paper would be easier to read if the comparison between study 1 and 2 was moved in the results section (now the first sentence of the discussion seems a bit unjustified at first because we haven't seen the results at that point)

Referee: 2

Comments to the Author(s)

Review: Great Apes and Human Children Rationally Monitor Their Decisions

Comments to the Authors:

The authors present a novel and very interesting way of testing apes' & young children's ability to seek information after receiving conflicting information. The design well thought out and is a good test of the question at hand. This manuscript would have broad appeal in comparative cognition, developmental psychology, and cognitive psychology. The authors conclude that apes seek information after receiving conflicting information directly, but not when the conflicting information is given via a social partner. In contrast, three-year-olds looked more when the competing information is from a social partner. The manuscript is well written and clear. The one large issue I have is that the statistics used do not directly test the conclusions made in the paper (see below). However, given the graphs in the supplemental, I think the data is there and will likely lead to the same conclusions. With some major revisions, the paper would be a great fit for Proc. B.

1. The term "Rational Decision Making" seems a bit of a stretch given the evidence. There seems like two reasonable possibilities for how the information from different sources is combined.

1) "comparing those competing piece of evidence or 'reasons' for making the decision against one another", the reason the authors presume.

2) An alternative is that the animals & children put some type of weight on the information they receive immediately and then just compare these weights. There would be no need to think about where each of the evidence comes from or to do any type of “reason-based” monitoring once the evidence has been given a weight.

While this seems like a small point, it is at the heart of the argument in the paper would benefit from further clarification whether the authors think possibility #2 is still rational (reason-based) or not.

2. Study 1 analysis: The method of using a GLMM works well for one of the conclusions drawn from the analyses (that collapsed across all groups there was an effect of condition overall effect of condition across all groups). However, it is not fit for some of the conclusions made (e.g. which group is driving the overall effect of condition). The finding that there was no interaction between population and condition exemplifies this issue. To then go on and say that the effect is driven by any subset of these groups (e.g. apes/5yos) is unfounded given the statistics presented.

- a. There are two reasonable ways to approach this: The first is to use contrast coding within the larger model to look at specific differences between the groups, the other is to run three separate, simpler models on each of the groups.
- b. In the full model was “condition” significant before removing all of the included variables?
- c. Given that the data is clearly not normally distributed, graphing the means for each group/condition does not seem like the most informative graph. A violin plot would highlight the wide variety of responses that were given. OR the type of plots used in figure S1/S2.
- i. Furthermore, on this point a statement about how many animals and children are driving these effects seems necessary. It is clearly not the case that all of the animals look when they get conflicting information (at least 9/18 monkeys, 15/66 5yos, & 18/64 3yos never looked)
 1. How many children/monkeys looked more often in the conflicting case?
 2. How many actually looked more in the consistent case?
- d. Given the above questions about the data, the conclusions that “Great apes can detect their own uncertainty... when the reasons for their decision are in conflict.” and “This ability has emerged by age five in human children”, seems like a stretch given that at least 15 5yos never looked and half of the apes never looked. Reporting how many subjects in each group looked more often (compared to how many looked less often) when there was conflicting info is necessary for these conclusions.

3. Study 2 analysis: My comments about the study 1 analysis all apply to this analysis as well. The only conclusion that can be made given the statistics used is that when all subjects (regardless of group) are collapsed, there is an effect of condition

- a. Here the conclusion that 3yos are now successful is unfounded given the statistics used (as is the finding that apes failed).
- b. All of the groups seem to have a bias to not look in experiment 1 (regardless of condition), and a bias to look in experiment 2.
 - i. What is the reason for this?
 - ii. Why is the effect size between the difference tasks (social/not social) much larger than any of the effect seen in condition (consistent/not consistent)?

4. General Discussion

- a. The authors address the differences between the specific groups, but do so in a roundabout way that does not completely fix the issues raised in the analysis sections.
 - i. The statistic used shows there was an interaction effect between experiment & condition in apes only shows that they peeked more in the physical condition compared to the social condition. A better way to do this would be to compare each of these to chance, i.e. is there a significant effect of condition in the physical experiment for apes? Is there a difference between conditions in the social experiment?
 1. The above also applies for the conclusion about 3yos.
 - ii. Given that these findings are the base of the entire study, more direct analyses are needed.

b. The explanation for why 3-year-olds are better in the social condition is unclear. It is not the case that experiment 2 was a 100% social task. It was a combination of a social and a physical task. Subjects did not just need to pay attention to social cues, but to the direct physical evidence as well (in which they failed to do in experiment 1). How/why do children pass on the version that integrates social and physical evidence when they fail on physical evidence alone. The explanation that they are better in social-contexts seems incomplete.

Minor comments:

- Paragraph 2/line 1: There has also been work showing that monkeys can tell when they have some, but not enough information (see Brady & Hampton, 2021), this is beyond know they have no information.
- Were there any differences between ape species?
- SM/methods & materials: Were the subjects that were excluded counted in the subject count in SM:Participants and the main text? I assume not, but a clarification would be good.
- What box did the subjects choose when they did not look in the inconsistent trials?
- Did they choose at random? Or did they choose the most recent information, e.g. the box that showed the item to be larger after the turn?
- Table S1. Seems uninformative as is. Instead of broad conclusions, including the direct statistics in these boxes would be much better. E.g. each of the boxes represent the results of a model using "looking" as the dependent variable and condition as the predictor. That way you have 6 models each summarized in a box rather than a conclusion without any stats to back it up.

References:

Brady, R. J., & Hampton, R. R. (2021). Rhesus monkeys (*Macaca mulatta*) monitor evolving decisions to control adaptive information seeking. *Animal Cognition*, 1-9.

-Stephen Ferrigno

Referee: 3

Comments to the Author(s)

The paper presents some clever manipulations that reveal some interesting aspects of human and nonhuman information-seeking behavior. But, it is seriously marred by an unnerving and unjustified predilection toward over-interpretation and using mentalistic notions when simpler behavioral interpretations would suffice. Many terms and expressions are set in quotation marks leaving the reader to wonder if these are scientifically suitable notions or just window dressing to attract greater public attention.

The paper would definitely profit by the authors taking a more circumspect approach and more seriously considering the manipulations and findings from a behavioral perspective. After all, we only have what can be observed and confirmed--the rest is sheer speculation.

Furthermore, the paper is not well written. There are a few too many grammatical and punctuation errors as well as a serious mislabeling in the caption of Fig. S1. More careful preparation of a revision would certainly be in order.

Finally, there is something of an unholy blending of within-subject and between-subject manipulations finding their way into the same statistical analyses. There is also concern that the two studies might not properly be included in a single statistical analysis. And, there are very few individuals from some ape species in the studies and their data are also quite disparate; can they all legitimately be lumped into a single 'ape' group?

Author's Response to Decision Letter for (RSPB-2021-1076.R0)

See Appendix A.

RSPB-2021-2686.R0

Review form: Reviewer 1

Recommendation

Accept with minor revision (please list in comments)

Scientific importance: Is the manuscript an original and important contribution to its field?

Good

General interest: Is the paper of sufficient general interest?

Excellent

Quality of the paper: Is the overall quality of the paper suitable?

Good

Is the length of the paper justified?

No

Should the paper be seen by a specialist statistical reviewer?

No

Do you have any concerns about statistical analyses in this paper? If so, please specify them explicitly in your report.

No

It is a condition of publication that authors make their supporting data, code and materials available - either as supplementary material or hosted in an external repository. Please rate, if applicable, the supporting data on the following criteria.

Is it accessible?

Yes

Is it clear?

Yes

Is it adequate?

Yes

Do you have any ethical concerns with this paper?

No

Comments to the Author

I thank the reviewers for addressing my comments and providing new analysis and a careful examination of potential alternative interpretations, the manuscript has greatly improved in my opinion.

The new analysis reporting where participants look after seeing conflicting new evidence is great! I believe this is now the first study that is able to fully address the 2 concerns I raised during the first round of review in both children and apes simultaneously... This has been a pervasive problem in developmental and comparative studies of metacognition for quite some time, so perhaps this aspect could be emphasized more in the discussion by dedicating a paragraph to this issue, and referring to past findings in both children & apes?

I still have one comment about some of the interpretation of the findings. This is perhaps only related to wording and should be easy to fix:

“apes were more sensitive to conflicting physical evidence than social evidence”

This is a strong conclusion, and I am still wondering about one slightly alternative interpretation: is it possible that the cross-species difference is due to more general differences in the way individuals see other individuals by default (e.g., as cooperative or potentially deceitful)? Could it be that children see one another as genuine cooperative “partners” by default, while apes see other agents as potential competitors? “In this case withholding judgment and looking for more information requires the ability to understand not only that our own beliefs and the information it was based on might be false, but also that our partner’s belief may be false”: AND it requires evaluating the intentions of the other agent (i.e., is the partner likely to disclose information truthfully or not)? Would this interpretation in terms of social status/relationships not be more parsimonious, as it does not imply that apes would never consider conflicting social information, but rather, that they are not necessarily going to believe what a social “partner”’s behavior discloses about the location of a reward? Would this not be compatible with what we believe about apes’ social cognition and some of the last authors’ claims about the unique pro-sociality of humans? I can see how it may be argued that this is actually less parsimonious as it assumes that apes have the ability to attribute deceitful intents to the other agent? In any case, it would be good to discuss this.

Minor

p2.: I am still not super convinced that the current study targets a different competence as compared to past studies in the field of metacognition, i.e., that weighing uncertainty during decision making vs. revising beliefs in light of new conflicting evidence must involve strictly different processes (e.g., see work showing a continuity between the mechanisms supporting changes of mind in light of dynamically evolving evidence and confidence judgments, e.g., by Kiani, Shadlen and colleagues, or Fleming and colleagues etc...). One way to clarify this would be to refer to the distinction between prospectively evaluating competing options and confidence before making a choice, and retrospectively using past-choice/evidence to make subsequent decisions (i.e., prospective vs. retrospective metacognition). It is also the case that if you have to make a choice between an option A or B - for instance in the tone example - the tone can only be A or B, so there is a conflict / an incompatibility there too...

p.6: new description of the 3-way interaction, thanks this is really useful. Could you also provide the stats for the main effect of condition here for both the apes and the 3yo separately for each study?

Thanks for revising the description of the statistics. This is clearer, but still not 100% clear, perhaps just include the formulas in the methods and/or SM for readers to check this if they are interested? and or simply mention that the formulas are available on OSF somewhere in the description of the stats.

“yes, children are better at monitoring their own knowledge and recognize that they lack information in the social study - just as you have suggested »

Yes, but a lot of the directly relevant literature on the development of metacognition is still not covered in the paper to reflect the fact that we already thought this was the case; e.g., studies by Kim, Sodian, Proust and colleagues have already shown that young children’s metacognition is potentiated in social contexts...

The discussion is really short and - also see above - it would be nice to expand a bit on some aspects. e.g., "This fits with theoretical models that take human metacognition to be adapted for solving peer disputes (31), (but the claim is somewhat wider than this...? i.e., metacognition has a social function, not simply for solving conflicts, but also other purposes like knowledge transmission etc., for instance see also papers by Heyes, or Dunstone and Caldwell...) and that human rationality is originally social (19-21)."

I don't know if the findings fit so comfortably with this, as they also show that reasoning is not typically human...? So how does that relate to claims (e.g., by Sperber/Mercier or others that are already cited in the paper) that reasoning evolved to fulfill a social function? It could be good to expand a little bit on this.

"but are instead engaged in involuntary information-seeking triggered by a cue" do you mean automatic rather than involuntary?

Review form: Reviewer 2

Recommendation

Accept with minor revision (please list in comments)

Scientific importance: Is the manuscript an original and important contribution to its field?

Excellent

General interest: Is the paper of sufficient general interest?

Excellent

Quality of the paper: Is the overall quality of the paper suitable?

Excellent

Is the length of the paper justified?

Yes

Should the paper be seen by a specialist statistical reviewer?

Yes

Do you have any concerns about statistical analyses in this paper? If so, please specify them explicitly in your report.

Yes

It is a condition of publication that authors make their supporting data, code and materials available - either as supplementary material or hosted in an external repository. Please rate, if applicable, the supporting data on the following criteria.

Is it accessible?

Yes

Is it clear?

Yes

Is it adequate?

Yes

Do you have any ethical concerns with this paper?

No

Comments to the Author

The authors have done a great job addressing my concerns and I think the manuscript should be published. However, I do still have one minor comment about the statistics used to draw conclusions about success and failure within each study for a given population.

My concern is exemplified in Table S1. The statistics used here (the interaction between condition & study for each population) is the correct statistic to show that there was a difference between conditions and rightly justifies the conclusions that apes are more sensitive to physical than social cues, and 3-year-olds are more sensitive to social than physical cues. However, there is no direct tests of success or failure on the individual studies (e.g. the "Yes" and "No" that is included in the table). The only statistics used within a study show that collapsed across all groups, there is a significant effect of condition in both Exp. 1 & Exp. 2. This does not speak to success or failure of individual populations within a group.

The two passages below make conclusions about results from specific populations within a study that currently lack statistical evidence:

"Great apes can detect their own uncertainty not only when they have no or ambiguous information (8- 11), but when the reasons for their decision are in conflict" pg.4 line 12-14,

"The results of our second study indicate that children as young as three take peer-disagreement as a reason to call their beliefs into question." pg.6 line 8-9

If the authors want to make these claims, I suggest they run pairwise comparisons between conditions for each study (e.g. testing if the effect of condition in Study 1 is significant and likewise with Study 2). This can be done using the same regression already being implemented on page 6 & SI "Studies 1 and 2 compared" for separate populations. Alternatively, the passages noted above (and the "Yes"/"No" in Table S1) could be removed.

I would like to note that a majority of the conclusions drawn are distinctly about the differences between Study 1 & 2 (for the different populations) which has been clearly established by the authors using the interaction of study*condition for each population. These conclusions are well warranted given the data and analyses used.

Decision letter (RSPB-2021-2686.R0)

24-Jan-2022

Dear Professor O'Madagain:

Your manuscript has now been peer reviewed and the reviews have been assessed by an Associate Editor. The reviewers' comments (not including confidential comments to the Editor) and the comments from the Associate Editor are included at the end of this email for your reference. As you will see, the reviewers and the Editors have raised some concerns with your manuscript and we would like to invite you to revise your manuscript to address them.

To submit your revision please log into <http://mc.manuscriptcentral.com/prsb> and enter your Author Centre, where you will find your manuscript title listed under "Manuscripts with

Decisions." Under "Actions", click on "Create a Revision". Your manuscript number has been appended to denote a revision.

Research ethics:

Use of animals and field studies:

It is a condition of publication that you make available the data and research materials supporting the results in the article (<https://royalsociety.org/journals/authors/author-guidelines/#data>). Datasets should be deposited in an appropriate publicly available repository and details of the associated accession number, link or DOI to the datasets must be included in the Data Accessibility section of the article (<https://royalsociety.org/journals/ethics-policies/data-sharing-mining/>). Reference(s) to datasets should also be included in the reference list of the article with DOIs (where available).

All supplementary materials accompanying an accepted article will be treated as in their final form. They will be published alongside the paper on the journal website and posted on the online figshare repository. Files on figshare will be made available approximately one week before the

accompanying article so that the supplementary material can be attributed a unique DOI. Please try to submit all supplementary material as a single file.

Please submit a copy of your revised paper within three weeks. If we do not hear from you within this time your manuscript will be rejected. If you are unable to meet this deadline please let us know as soon as possible, as we may be able to grant a short extension.

Best wishes,
Dr Robert Barton
mailto:proceedingsb@royalsociety.org

Associate Editor

Comments to Author:

This is a revision of a paper examine the evolutionary and developmental origins of metacognitive reasoning, implementing a novel task to test what kinds of information children and apes use to assess what they know. The reviewers and I agree that this is a responsive revision that has greatly improved the paper. However, the reviewers also have several helpful comments on new version of the manuscript that should still be clarified. As an additional point, I agree that the new coding and analysis of location of searching is a nice addition, but I think this primary reporting of the coding and analyses of this measure needs to be moved to the methods and results section rather than reported for the first time in the general discussion.

Reviewer(s)' Comments to Author:

Referee: 1

Comments to the Author(s).

I thank the reviewers for addressing my comments and providing new analysis and a careful examination of potential alternative interpretations, the manuscript has greatly improved in my opinion.

The new analysis reporting where participants look after seeing conflicting new evidence is great! I believe this is now the first study that is able to fully address the 2 concerns I raised during the first round of review in both children and apes simultaneously... This has been a pervasive problem in developmental and comparative studies of metacognition for quite some time, so perhaps this aspect could be emphasized more in the discussion by dedicating a paragraph to this issue, and referring to past findings in both children & apes?

I still have one comment about some of the interpretation of the findings. This is perhaps only related to wording and should be easy to fix:

"apes were more sensitive to conflicting physical evidence than social evidence"

This is a strong conclusion, and I am still wondering about one slightly alternative interpretation: is it possible that the cross-species difference is due to more general differences in the way individuals see other individuals by default (e.g., as cooperative or potentially deceitful)? Could it be that children see one another as genuine cooperative "partners" by default, while apes see other agents as potential competitors? "In this case withholding judgment and looking for more information requires the ability to understand not only that our own beliefs and the information it was based on might be false, but also that our partner's belief may be false": AND it requires evaluating the intentions of the other agent (i.e., is the partner likely to disclose information

truthfully or not)? Would this interpretation in terms of social status/relationships not be more parsimonious, as it does not imply that apes would never consider conflicting social information, but rather, that they are not necessarily going to believe what a social “partner”’s behavior discloses about the location of a reward? Would this not be compatible with what we believe about apes’ social cognition and some of the last authors’ claims about the unique pro-sociality of humans? I can see how it may be argued that this is actually less parsimonious as it assumes that apes have the ability to attribute deceitful intents to the other agent? In any case, it would be good to discuss this.

Minor

p2.: I am still not super convinced that the current study targets a different competence as compared to past studies in the field of metacognition, i.e., that weighing uncertainty during decision making vs. revising beliefs in light of new conflicting evidence must involve strictly different processes (e.g., see work showing a continuity between the mechanisms supporting changes of mind in light of dynamically evolving evidence and confidence judgments, e.g., by Kiani, Shadlen and colleagues, or Fleming and colleagues etc...). One way to clarify this would be to refer to the distinction between prospectively evaluating competing options and confidence before making a choice, and retrospectively using past-choice/evidence to make subsequent decisions (i.e., prospective vs. retrospective metacognition). It is also the case that if you have to make a choice between an option A or B – for instance in the tone example - the tone can only be A or B, so there is a conflict / an incompatibility there too...

p.6: new description of the 3-way interaction, thanks this is really useful. Could you also provide the stats for the main effect of condition here for both the apes and the 3yo separately for each study?

Thanks for revising the description of the statistics. This is clearer, but still not 100% clear, perhaps just include the formulas in the methods and/or SM for readers to check this if they are interested? and or simply mention that the formulas are available on OSF somewhere in the description of the stats.

“yes, children are better at monitoring their own knowledge and recognize that they lack information in the social study – just as you have suggested »

Yes, but a lot of the directly relevant literature on the development of metacognition is still not covered in the paper to reflect the fact that we already thought this was the case; e.g., studies by Kim, Sodian, Proust and colleagues have already shown that young children’s metacognition is potentiated in social contexts...

The discussion is really short and - also see above - it would be nice to expand a bit on some aspects. e.g., “This fits with theoretical models that take human metacognition to be adapted for solving peer disputes (31), (but the claim is somewhat wider than this...? i.e., metacognition has a social function, not simply for solving conflicts, but also other purposes like knowledge transmission etc., for instance see also papers by Heyes, or Dunstone and Caldwell...) and that human rationality is originally social (19-21).”

I don’t know if the findings fit so comfortably with this, as they also show that reasoning is not typically human...? So how does that relate to claims (e.g., by Sperber/Mercier or others that are already cited in the paper) that reasoning evolved to fulfill a social function? It could be good to expand a little bit on this.

“but are instead engaged in involuntary information-seeking triggered by a cue” do you mean automatic rather than involuntary?

Referee: 2

Comments to the Author(s).

The authors have done a great job addressing my concerns and I think the manuscript should be published. However, I do still have one minor comment about the statistics used to draw conclusions about success and failure within each study for a given population.

My concern is exemplified in Table S1. The statistics used here (the interaction between condition & study for each population) is the correct statistic to show that there was a difference between conditions and rightly justifies the conclusions that apes are more sensitive to physical than social cues, and 3-year-olds are more sensitive to social than physical cues. However, there is no direct tests of success or failure on the individual studies (e.g. the "Yes" and "No" that is included in the table). The only statistics used within a study show that collapsed across all groups, there is a significant effect of condition in both Exp. 1 & Exp. 2. This does not speak to success or failure of individual populations within a group.

The two passages below make conclusions about results from specific populations within a study that currently lack statistical evidence:

"Great apes can detect their own uncertainty not only when they have no or ambiguous information (8- 11), but when the reasons for their decision are in conflict" pg.4 line 12-14,

"The results of our second study indicate that children as young as three take peer-disagreement as a reason to call their beliefs into question." pg.6 line 8-9

If the authors want to make these claims, I suggest they run pairwise comparisons between conditions for each study (e.g. testing if the effect of condition in Study 1 is significant and likewise with Study 2). This can be done using the same regression already being implemented on page 6 & SI "Studies 1 and 2 compared" for separate populations. Alternatively, the passages noted above (and the "Yes"/"No" in Table S1) could be removed.

I would like to note that a majority of the conclusions drawn are distinctly about the differences between Study 1 & 2 (for the different populations) which has been clearly established by the authors using the interaction of study*condition for each population. These conclusions are well warranted given the data and analyses used.

Author's Response to Decision Letter for (RSPB-2021-2686.R0)

See Appendix B.

Decision letter (RSPB-2021-2686.R1)

28-Feb-2022

Dear Professor O'Madagain

I am pleased to inform you that your manuscript entitled "Great Apes and Human Children Rationally Monitor Their Decisions" has been accepted for publication in Proceedings B.

Data Accessibility section

Open Access

Your article has been estimated as being 6 pages long. Our Production Office will be able to confirm the exact length at proof stage.

Paper charges

Sincerely,

Dr Robert Barton

Associate Editor:

Comments to Author:

The revision does a nice job addressing the remaining comments. I agree that referencing the ignorance-knowledge test in the main manuscript is helpful.

Appendix A

Replies to Reviewers and Associate Editor

Associate Editor: pages 2-4

Referee 1: pages 5-13

Referee 2: pages 14-22

Referee 3: pages 23-26

Note from authors:

Sincere thanks to all the associate editor and all three reviewers for their time and the thoughtful challenges they have raised, we think the paper has been substantially improved by the revisions.

Associate Editor
Comments to Author:

This is a study of metacognition in apes and children. Understanding of metacognitive information seeking has emerged as an important cognitive skill with repercussions for many aspects of mature human behaviors, and this set of studies develop a clever way to address its developmental and phylogenetic roots, and further introduces a novel distinction about the source of conflicting information (social versus physical) that may then lead to information-seeking. The reviewers (and I) agree that this is an interesting topic and novel experimental paradigm, but also have helpful comments regarding the paper that should be addressed in revision.

Thank you for your encouraging remarks, we are very happy that you find our work interesting.

One important point concerns the statistics and interpretation of the results. For example, R1 and R2 point out several comments about how the current statistical reporting does not always support the conclusions drawn from the studies, with the most salient being that the main outcome of Study 2 is that there is a main effect of condition (and no improvement by including the interaction with population) but the interpretation is written as though there was a significant interaction.

Thank you. We agree that the models that look at each study separately do not support the claim that there are differences in the ways in which different populations respond to the studies; however, the larger model that includes both studies reveals significant interactions between study and condition in both the ape and three-year old groups, supporting the claims. These interactions show that the apes distinguish the conditions in the individual study but not the social study, while the three-year old children do the opposite. This analysis therefore supports the claims about distinctions between groups.

We believe it was an error on our part not to make the centrality of this analysis clear enough – as reviewer 1 points out (in their final remark), the claims we make are indeed supported by the final analysis but it was not in the best location in the paper. We have therefore moved that analysis into the results section to make its centrality to our claims clearer (MM, 6, line 15ff):

“Overall we had predicted that great apes would be more sensitive to a conflict in physical rather than social information, while young children would be more sensitive to conflicting social information. To test this directly we pooled all data and tested for a three-way interaction of population, condition, and study – to see if different groups were more sensitive to the distinction between conditions from one study to the other. This was significant ($\chi^2=11.408$, $df=2$, $P=0.003$). To isolate the source of the three-way interaction we tested for an interaction between study and condition in each population separately. We found an interaction between condition and study in the apes ($\chi^2=4.312$, $df=1$, $P=0.037$). This shows us that the apes distinguished the conditions more in the physical study than in the social study, or, that apes are more sensitive to contradictory physical information than contradictory social information. We also found an interaction between study and condition in the three-year olds ($\chi^2=11.596$, $df=1$, $P<0.001$). This time the effect is reversed: the three year old children distinguished the conditions more

in the social study than in the physical study. There was no interaction in the five-year olds, but rather a main effect of condition: the five-year olds peeked significantly more when faced with conflicting evidence in both studies ($\chi^2=13.746$, $df=1$, $P<0.001$) (see Table S1 for an overview). These results support our basic hypothesis: apes were more sensitive to conflicting physical evidence than social evidence, while the youngest children were more sensitive to conflicting social than physical evidence.”

R1 and R3 also comment about the particularities of how these populations (younger and older children versus apes) are grouped in analyses, and the paper would benefit from more clearly explaining the logic here.

A great deal of research shows that cognition changes significantly in human development between three and five years, which is why these age groups are often compared in developmental psychology. This is particularly the case concerning children’s developing understanding of belief – which is of central importance to the current study (refs 15-18). On some views it is at the older of these two ages that children acquire cognitive skills that really set us apart from other animals, particularly those that depend on language or enculturation. We have clarified this point in the main manuscript (MM, 2, line 45):

Participating in our first study were apes (N=18), three-year old (N=64) and five-year old children (N=66). We chose these age groups because a good deal of evidence indicates that children’s understanding of belief develops substantially between three and five years (15-18), so that by including both we could identify any developmental change that takes place.

The reviewers have several other helpful comments about statistical approaches and best practices for reporting that should be addressed. Better clarity on the use of GLMMs will likely also address comments from R3 about within- versus between-subjects manipulation

Thank you. We have addressed reviewer 2 and 3’s comments on our statistical tests in detail below – in particular on our use of interactions to support our claims, which are the most direct tests for the claims we are making. We have also commented in the supplementary materials (SM6) on the way in which GLMM can be used to compare within and between subject data, as raised by reviewer 3.

Another important point is to provide a more robust theoretical framework for the study that better incorporate vies from cognitive science exploring the particular mechanisms that may underpin information seeking. R1 provides some helpful ideas, references and other possible interpretations that could be helpful to this end to provide a more nuanced interpretation of the actual cognitive processes underlying responses in the task. R2 also points out similar comments related to the use of the term ‘rational decision making.’ I believe a careful treatment of these issues will also help address R3’s concerns about mentalistic interpretation of the results.

Thank you. We have expanded our general discussion substantially to address concerns of all three reviewers. Particularly, we conducted an additional analysis of the data to address a challenge of R1 and R2 – that perhaps participants are only considering the evidence ‘sequentially’ (following Leahy and Carey 2020). On this view, participants would not be metacognitively considering the prior belief along

with the new evidence, but would only be thinking of the new evidence when it appears.

We considered that if this were true, then it should have an impact on *where* participants look when they double-check the evidence. The new evidence is identical in both conditions, after all. Therefore, if they are only thinking of the new evidence, they should be just as likely to double-check in the location indicated by the new evidence in both conditions. To investigate this we recoded the videos to check where participants looked when they sought more information. As you can see from figure S5, participants are far more likely to check in the *opposite* location to that indicated by the new evidence in the ‘conflicting’ condition. This is difficult to explain if participants are only thinking of the new evidence, and provides elegant new support for our interpretation – that participants double-check because they are still thinking of their earlier belief.

We have explored these objections and presented our replies in the expanded general discussion (MM, 7, line 18ff):

“One challenge to our interpretation of this behavior is to suppose that participants are perhaps considering just one piece of evidence at a time, ‘sequentially’, rather than reconsidering their prior belief in light of the new information (27). On this view, participants would only be thinking of the new evidence when they look for more information. To address this concern, we looked at which box participants peeked in first, when they checked for more information. We supposed that if participants are only considering the most recent piece of evidence, then they should be just as likely to peek in the location indicated by the new evidence in both conditions – since the new evidence is identical in both conditions. However, we found that in the ‘Conflicting’ trials participants were significantly more likely to peek in the opposite location to that indicated by the new evidence than in the ‘Consistent’ trials ($\chi = 7.93$; $P < 0.0001$) (Figure S5). This is hard to explain if the participants are only thinking of the most recent evidence, and adds weight to our interpretation that the participants are indeed thinking of their prior belief and the new evidence at once.

Another concern that could be raised is that perhaps, when faced with conflicting information, an involuntary hesitation might become a cue that participants could use to learn that whenever they experience it, they will do better to look for more information before choosing (13). On this view participants would simply react to a ‘feeling of uncertainty’ that they use as a cue to information-seeking. Alternatively, we might worry that this behavior could be generated by a decision taking too long, triggering information-seeking as a sort of ‘reset’ to return to foraging (28). On both of these interpretations, participants do not know what they do not know, but are instead engaged in involuntary information-seeking triggered by a cue. Arguably if this were true we should expect that they would search randomly for food when their information-seeking began – instead of looking for exactly the information they need to answer the question they are uncertain about. The participants in the current studies, however, do not search at random; they look for exactly the information they need to resolve the conflict (peeking inside the containers they have already been exploring). This ‘targeted information-seeking’ (3) is best explained by supposing that participants recognize what it is they do not know, and are seeking the information they need to specifically address this question.”

.....

Reviewer(s)' Comments to Author:

Referee: 1

Comments to the Author(s)

The authors present an interesting study suggesting that there are differences in children and apes' ability to monitor their confidence and seek information when a peer acts in a direction that is in conflict with their own belief. I think study2 is interesting, but that there are a few alternative interpretations of the results at the moment, and further analysis could potentially allow the authors to rule them out. The manuscript also needs some work in order to better situate the results with respect to past research.

Major

« we do not know whether any nonhuman animals recognize when they cannot make a reliable decision because they have several conflicting pieces of information. » ; “what has not been investigated is whether young children or animals can detect their own uncertainty because they have conflicting information »

We do know a lot about this, e.g., see Hampton 2009; Kornell 2007; Kepecs et al., 2012; Rosati et Santos 2016... In many animal studies on metacognition, one could argue that there is ambiguity in the sensory evidence that forces the animal to weigh competing options... The authors argue that past research has looked at cases where no information is present, but there are actually a lot of perceptual decision studies where animals need to weigh 2 response options as a function of ambiguous sensory evidence (e.g., see reviews by Smith et al., 2012; Hampton 2009; Kepecs et al., 2012; Proust 2019...). Similarly, in children studies on metacognitive monitoring often rely on presenting ambiguous sensory evidence (e.g., see Lyons and Ghetti 2013; Geurten & Basin 2018; Beran et al., 2012; Baer and Odic 2021; Vo et al., 2014...). Arguably, when faced with a binary choice an agent will always weight competing options and conflicting evidence? The fact that a lot of the relevant literature on metacognition in children and animals is not covered seems problematic...

Thank you.

First let us try to capture here more fully what is new in these studies that goes beyond previous work:

In our study, participants received two successive pieces of evidence that contradict each other. This is not *ambiguous* evidence, it is contradictory – both apparent situations cannot be true simultaneously. First, evidence that the best reward is definitely on the left (let's say), which induces the participant to form a belief that she should choose the left, which she does. Second, evidence that the best reward is definitely on the right. Both of these pieces of information cannot be correct at once – one of them must be 'wrong'. To recognize the problem, the participants must therefore understand that their prior belief, or else the new information, must be wrong.

Nothing like this arises in the studies with ambiguous stimuli. Consider one version of the opt-out task (Crystal & Foote 2007). Here participants (rats) learn that if they hear a long tone, they get a reward if they press the lever on the left; and if they hear a short tone, they get a reward if they press a lever on the right. Then a medium length tone is heard that is neither short nor long, and therefore harder to classify – and participants learn to skip these trials. But at no point does the participant form a belief that s/he then has to call into question, or receive new evidence that contradicts prior evidence. The medium length tone does not ‘contradict’ the long or short tone. What is lacking in the ambiguous stimuli tasks is a requirement that the participant recognize that they might be making a mistake, or that their current evidence may be misleading. This is what sets our current studies apart.

Obviously we have not made this clear enough so we welcome the opportunity to review our opening to highlight these differences (Main manuscript, 2):

“These studies demonstrate the ability to detect uncertainty due to having ambiguous or insufficient information. While they may reveal that a subject can detect when they lack sufficient information, however, they do not reveal an ability to think about what one already believes, which is a different kind of metacognition. Thinking about what one believes is sometimes elicited when one encounters new evidence that calls an already-formed belief into question. In such a scenario, we compare the evidence or reason we had for forming our original belief with the new evidence or reason we have for revising that belief, recognizing that either one could be misleading. Based on the terminology used by many philosophers (14) it can therefore be called ‘rational’ or ‘reason-based’ monitoring of the decision-making process.”

Although we do already cite several of the papers mentioned above, we have expanded our discussion of previous metacognition studies including in the general discussion at the end.

It seems that the main novelty here concerns the findings about meta-reasoning (e.g., see Ackerman & Thompson 2017) in the face of social contradiction (i.e., study 2). Some reworking of the introduction/abstract seems required to clarify what is novel here (e.g., shifting the focus on the novel findings about social contradiction, and discuss at greater length the inter-species differences and what they may mean)?

We hope that our revised introduction, above, makes it clear that the novelty of the two studies concerns the contradictory evidence; but we have also expanded the point about the inter-species difference with respect to the social study:

“We expected this task would be less compelling for apes, however, whose cognition has evolved primarily for individual problem-solving (21). Finding such a contrast, we expected, would illustrate the primarily social nature of human cognition, as opposed to the primarily individual nature of ape cognition.”

It would also be important to discuss points raised e.g., by Hampton 2009; Perner and Dienes 2013; Leahy and Carey 2020..., that in this kind of information seeking task it is difficult to know whether agents engage in metacognitive monitoring, rather than trying options sequentially, and/or engaging in information seeking automatically when no response comes to mind. For instance, Hampton (2009, also see Perner & Dienes 2013) has suggested that

information seeking in this type of task could be driven by response competition, as a form of default option triggered when the decision-making fails after a certain period of time (i.e., a sort of “time is up” automatic response). Can the authors provide more data analysis (for instance an analysis of response times for peeking versus choosing, e.g., see Goupil et al., 2016) that could allow them to rule out this response competition interpretation? Another important point to address is the possibility that apes/children do not really weight conflicting information here, but instead estimate how probable 1 option is at a time (e.g., see Leahy and Carey 2020). Here nothing comes to my mind in terms of additional data analysis that could rule out this interpretation, but there may be some aspects in the data that speak against this possibility? If not some interpretations will have to be readjusted in function (especially the conclusions of the paper).

These are interesting challenges. First let us consider the challenge of Leahy and Carey (2020):

We think there is indeed a way to address it – by analyzing *where* participants check first when they are given the opportunity to check. We supposed that if participants were only thinking of the new evidence, then they should be just as likely to first check in the location indicated by the new evidence in both conditions. After all, the new evidence is identical in both conditions. Even if they peek more often in the ‘conflicting’ condition, there shouldn’t be a difference in *where* they peek between conditions, if they are only thinking of the new evidence whenever they get the opportunity to peek.

To explore this we recoded the videos and analyzed for the likelihood to first check the location the new evidence indicated. We found that there was a highly significant difference between conditions: participants were significantly more likely to check the opposite location to that indicated by the new evidence in the ‘conflicting’ trials than in the ‘consistent’ trials ($\chi = 7.93$; $P < 0.0001$). This provides helpful additional evidence that participants are indeed thinking of the prior belief along with the new evidence, since their prior belief indicates the opposite location to the new evidence in the conflicting condition (MM, 7):

“One challenge to our interpretation of this behavior is to suppose that participants are perhaps considering just one piece of evidence at a time, ‘sequentially’, rather than reconsidering their prior belief in light of the new information (27). On this view, participants would only be thinking of the new evidence when they look for more information. To address this concern, we looked at which box participants peeked in first, when they checked for more information. We supposed that if participants are only considering the most recent piece of evidence, then they should be just as likely to peek in the location indicated by the new evidence in both conditions – since the new evidence is identical in both conditions. However, we found that in the ‘Conflicting’ trials participants were significantly more likely to peek in the opposite location to that indicated by the new evidence than in the ‘consistent’ trials ($\chi = 7.93$; $P < 0.0001$) (Figure S5). This is hard to explain if the participants are only thinking of the most recent evidence, and adds weight to our interpretation that the participants are indeed thinking of their prior belief and the new evidence at once.”

The details are added to the supplementary materials, along with the following plot of results (SM9, Figure S5):

Figure S5.

Likelihood to check alternative location to new evidence first: Here we see the rate at which participants check the opposite location to that indicated by the new evidence. Boxes represent mean and standard error, bubble size indicates the number of trials on which the alternative location was checked first. It can be seen that in the ‘conflicting’ condition, participants are far more likely to check the opposite box to that indicated by the new evidence than they are in the ‘consistent’ condition. This is hard to explain if we suppose that participants are only thinking of the ‘new’ evidence, since the new evidence is identical in both conditions.

Now let us consider the challenge of Hampton (2009)/Perner and Dienes (2013):

This challenge concerns the possibility that participants engage in information-seeking as a sort of ‘time out’ response, which might be addressed by measuring the response times. Unfortunately the camera angles in the studies with apes make coding this timing impossible. However, we are not convinced that evidence that a peek takes longer than a choice would be evidence against metacognition. On the contrary, Suda and Call (2006), Smith, J. D., Shields, W. E., & Washburn, D. A. (2003), and William James (1890) argue that hesitation is evidence *in favor* of metacognition, or ‘secondary decision-making systems’:

“[Consider] the theory that cognitive conflict prompts hesitation, which is a manifestation of a secondary decision-making system as opposed to automatic responses (James, 1890/1981; Smith, Shields, & Washburn, 2003; Tolman, 1938, 1932/1967). According to this theory,

metacognitive animals should show more hesitant behavior as their internal mental conflict intensifies” (Suda and Call 2006, 54).

Even though the analysis is not possible, however, we also believe that the ‘targeted information seeking’ in which participants engage undermines the idea that they have reverted to a sort of default foraging state. Rather than searching at random for food, they peek in exactly the place required for them to resolve the conflict they are faced with. This observation has been used in the past to address another related behaviorist interpretation – Proust and Carruthers’ notion of a ‘feeling of uncertainty’. And so we have addressed both of these in the general discussion (MM, 7):

“Another concern that could be raised is that perhaps, when faced with conflicting information, an involuntary hesitation might become a cue that participants could use to learn that whenever they experience it, they will do better to look for more information before choosing (13). On this view participants would simply react to a ‘feeling of uncertainty’ that they use as a cue to information-seeking. Alternatively, we might worry that this behavior could be generated by a decision taking too long, triggering information-seeking as a sort of ‘reset’ to return to foraging (28). On both of these interpretations, participants do not know what they do not know, but are instead engaged in involuntary information-seeking triggered by a cue. Arguably if this were true we should expect that they would search randomly for food when their information-seeking began – instead of looking for exactly the information they need to answer the question they are uncertain about. The participants in the current studies, however, do not search at random; they look for exactly the information they need to resolve the conflict (peeking inside the containers they have already been exploring). This ‘targeted information-seeking’ (3) is best explained by supposing that participants recognize what it is they do not know, and are seeking the information they need to specifically address this question.”

« engaging in curiosity-driven information seeking can be explained simply by being more interested in objects that behave in novel ways »: how is this supposed to work without metacognitive monitoring...? If curiosity is a desire for information (i.e., excludes other types of foraging), and that children engage in curiosity-driven information seeking when they are presented with novel or conflicting information because they realize that this does not conform to their prior knowledge, or that they cannot make sense of it, how can this be explained without them registering at some level that this activity could lead to a cognitive gain because they lack information? (i.e., without hypothesizing that there is a basic form of metacognitive monitoring involved); this point seems to contradict the rest of the argument upon which the paper is based... making a distinction between metarepresentations / explicit metacognition versus more basic forms of metacognition that do not necessarily require full-fledged, conceptual metarepresentations could be one way to go here (e.g., see Proust 2007; 2012; Shea et al., 2014; Goupil et al., 2019...). Another way to go is to not enter this debate about what curiosity requires at all.

We appreciate the suggestion to avoid the topic of curiosity and agree that it is tangential; since space is limited, we have removed the reference to the curiosity-based studies. We also moved the discussion of the studies with children to the general discussion at the end of the paper, to make the opening clearer. We think the opening is now much clearer overall.

The results are interesting in relation to the suggestion that explicit metacognition has a social function (Shea et al., 2014) and it could be nice to discuss the inter-species differences/similarities in this respect?

This is a nice suggestion. We have adjusted the final sentences as follows (MM, 7) (31 is Shea et al (2014)):

“This fits with theoretical models that take human metacognition to be adapted for solving peer disputes (31), and that human rationality is originally social (19-21). Our findings show that humans are not just *good* at problem-solving in social contexts, but *better* at solving problems socially than individually – and that the distinguishing mark of human cognition is its sociality.”

Study 1:

“This ability has emerged by age five in human children, but not at three”: it would be good to show the performances of the children and apes in choosing; where 3yo worse/slower? this could explain the difference in peeking in this age group... (i.e., a difference driven by task performances and/or motivation rather than metacognition) ; as mentioned above previous research suggests that 3yo can monitor their ability to discriminate between 2 options, but only when they are competent with the task at hand. A possible interpretation here is that 3yo were not competent enough in the discrimination task, and/or that they were not motivated enough to peek.

All participants who were included in the analysis passed an initial test to ensure that they would spontaneously peek when they had no information at all. In the case of children this was four preliminary trials where they see the boxes but get no view inside through windows – so they have no idea where the reward is. In these trials, children who did not peek inside spontaneously in 3 out of 4 trials were excluded. This was done to rule out the possibility that participants were not motivated enough to peek. See SM, 2, line 20. And so we think that this rules out the worry that the three year olds were not competent with the task.

Figure S4 suggests that in fact 3yo peek a lot, which suggests low confidence, where in general younger children have a bias of overconfidence when task performances are controlled for... So perhaps this suggests that 3yo are not so competent in this task (while ruling out the possibility that they are less motivated to peek overall).

Again the participants all passed the same entry requirements to show that they knew they could peek when they had no information, so we do not think a high amount of peeking explains the apparent failure of 3 year olds here to distinguish the conditions – note also that the participants are far from the ceiling. Generally all groups peeked more in the ‘social’ study, which we think is explained by the increased engagement with the task due to the presence of the social partner.

Other studies (some mentioned above) also suggest that young children may be better at monitoring their own knowledge and refrain from responding when they lack information when they are engaged in a social interaction... More discussion on this seems required.

Thank you, yes, children are better at monitoring their own knowledge and recognize that they lack information in the social study – just as you have suggested. This is

why the 3 year olds distinguish the conditions (conflicting/consistent) more in the social than the individual study (model 3). We have emphasized the connection between our results and the proposal of Shea et al 2014, as mentioned above.

Study 2: the interaction is not significant between apes, 3 and 5yo so it seems difficult to interpret the inter-species difference... was there an interaction if testing an effect of population considering only apes vs. children? Relatedly, would it not make sense to use population (children, apes) as a factor, and then test age effects by using only children data throughout the paper?

Thank you. There was not a significant interaction between populations when each study was considered separately, but when the studies are compared the interactions are significant (the final model reported). In other words, the degree to which different populations distinguished the conditions varied between studies; this was what we had hypothesized, that apes would distinguish the conditions more in the individual than in the social study, and that young children would distinguish the conditions more in the social study than in the individual study.

We found an interaction between study and condition for apes (they distinguish the conditions more in the individual than social study) and between study and condition for three year olds (the reverse effect - they distinguish the conditions more in the social than the individual study).

We think that we did not make the centrality of this crucial analysis clear enough in the main manuscript, and so we have revised this section (MM, 6, line 15ff):

“Overall we had predicted that great apes would be more sensitive to a conflict in physical rather than social information, while young children would be more sensitive to conflicting social information. To test this directly we pooled all data and tested for a three-way interaction of population, condition, and study – to see if different groups were more sensitive to the distinction between conditions from one study to the other. This was significant ($\chi^2=11.408$, $df=2$, $P=0.003$). To isolate the source of the three-way interaction we tested for an interaction between study and condition in each population separately. We found an interaction between condition and study in the apes ($\chi^2=4.312$, $df=1$, $P=0.037$). This shows us that the apes distinguished the conditions more in the physical study than in the social study, or, that apes are more sensitive to contradictory physical information than contradictory social information. We also found an interaction between study and condition in the three-year olds ($\chi^2=11.596$, $df=1$, $P<0.001$). This time the effect is reversed: the three year old children distinguished the conditions more in the social study than in the physical study. There was no interaction in the five-year olds, but rather a main effect of condition: the five-year olds peeked significantly more when faced with conflicting evidence in both studies ($\chi^2=13.746$, $df=1$, $P<0.001$) (see Table S1 for an overview). These results support our basic hypothesis: apes were more sensitive to conflicting physical evidence than social evidence, while the youngest children were more sensitive to conflicting social than physical evidence..”

Minor

Study 1: I am guessing the format of the journal prevents the authors from describing the paradigm at greater length, but still a lot of useful information seems to be missing from the description of the set-up on p.4 (e.g., how were participants shown that they could peek when

seeing the second display but not before? how many trials? what is “order”? how was peeking coded...) ; quite a lot of the information that is now in the SM seems required to allow readers to assess the findings.

Thank you. We have added some further explanations (MM, 3, line 5ff):

“To ensure that participants in the sample understood that they could check for more information, we ran a series of ‘warm-up’ trials, in which participants could not see the rewards through windows in the boxes, and could only make a decision by peeking over the top. Participants who did not peek spontaneously in these warm-up trials were excluded. Peeking was coded as a participant leaning forward to see inside the boxes; a second coder blind to the hypothesis replicated the coding to ensure reliability. Children received 3 trials in each condition, apes received 12. The order in which the conditions were presented to participants was pseudo-randomized to prevent participants learning to check for more information, as they might if ‘conflicting’ trials were presented in blocks.”

I am not sure I understand the model comparisons, was there no significant interaction between condition and population when comparing nested models like these:

peeking ~ condition + population

peeking ~ condition * population

Please clarify, this is ambiguous at the moment...

That’s right, there was no significant effect when comparing models with or without the interaction in the ‘within study’ analyses; however there were clear interactions in the between-study analyses. Our revision of the results section has hopefully made this clearer (also, we realized that the models as we presented them were a little unclear so we have simplified them; this simplification makes no difference to the conclusion but makes the model easier to interpret). The comparison you are referring to is:

```
# red0: peek ~ population + condition + control.mags + trial.m + reward.m + sex.m +  
(1 | subject) + (0 + trial.m | subject)
```

```
# full1: peek ~ population * condition + control.mags + trial.m + reward.m + sex.m +  
(1 | subject) + (0 + trial.m | subject)
```

Our models are all available for inspection on our OSF page here:

<https://osf.io/ey5f9/>

It may be easier to present the results of sequentially nested models (e.g., comparing the null model with a model including only condition, then adding population, then the interaction term or something like this).

These are, indeed, the results of sequentially nested models. There are two approaches to model comparison – starting with a small model and building up, or starting with a large model and reducing – we have employed the second strategy, which is thought to reduce the likelihood of type II errors since it allows all potential confounding factors to be included (rather than adding them one by one) (Forstmeier W & Schielzeth H. 2011. Cryptic multiple hypotheses testing in linear models: overestimated effect sizes and the winner’s curse. *Behav. Ecol. Sociobiol.*, 65, 47–55.)

p.5: “due to a surreptitious manipulation” what is this manipulation? it would be useful to describe this (e.g., in the Figure caption)

Thank you, we have added an explanation to the Figure caption 3 (MM, 5, line 5ff):

“In the ‘conflicting’ condition, the experimenter inserts the reward into one box, but then deposits it in a hole in the floor of the box; the other box had been ‘pre-loaded’ with an identical reward, so that when the non-target participant gets a view of the inside of the boxes, she sees the reward in the opposite location from where the experimenter appeared to have put it.”

p.7: the paper would be easier to read if the comparison between study 1 and 2 was moved in the results section (now the first sentence of the discussion seems a bit unjustified at first because we haven’t seen the results at that point)

Thank you this is a very helpful suggestion, we have done this as described above and feel the manuscript and results are much clearer now – see the passage quoted in full above (MM, 6, line 15ff).

.....
Referee: 2

Comments to the Author(s)

Review: Great Apes and Human Children Rationally Monitor Their Decisions

Comments to the Authors:

The authors present a novel and very interesting way of testing apes' & young children's ability to seek information after receiving conflicting information. The design well thought out and is a good test of the question at hand. This manuscript would have broad appeal in comparative cognition, developmental psychology, and cognitive psychology. The authors conclude that apes seek information after receiving conflicting information directly, but not when the conflicting information is given via a social partner. In contrast, three-year-olds looked more when the competing information is from a social partner. The manuscript is well written and clear. The one large issue I have is that the statistics used do not directly test the conclusions made in the paper (see below). However, given the graphs in the supplemental, I think the data is there and will likely lead to the same conclusions. With some major revisions, the paper would be a great fit for Proc. B.

1. The term "Rational Decision Making" seems a bit of a stretch given the evidence. There seems like two reasonable possibilities for how the information from different sources is combined.

1) "comparing those competing piece of evidence or 'reasons' for making the decision against one another", the reason the authors presume.

2) An alternative is that the animals & children put some type of weight on the information they receive immediately and then just compare these weights. There would be no need to think about where each of the evidence comes from or to do any type of "reason-based" monitoring once the evidence has been given a weight.

While this seems like a small point, it is at the heart of the argument in the paper would benefit from further clarification whether the authors think possibility #2 is still rational (reason-based) or not.

Thank you. Your proposal is similar to that of Leahy and Carey (2020) (also discussed in response to reviewer 1 who makes a similar point) who argue that in such a scenario, participants may be considering each piece of evidence 'sequentially', rather than thinking simultaneously of the old and the new evidence, under different weights. Since they are approaching the new evidence under a different prior in the 'conflicting' condition, they might peek more – but not because they are thinking of the prior belief, instead because their priors are simply different. We agree that this would not be metacognition as we had characterized it, where one thinks specifically of one's past belief when confronted with new evidence that undermines it.

To address this concern, we did a further analysis of the data. We considered that if participants were not thinking of the previous evidence, but only of the new evidence, then they should be just as likely to double the check the location indicated by the new evidence across all conditions. The new evidence is, after all, identical in both conditions. However, we found that in the 'conflicting' condition participants were significantly more likely to check in the opposite location to that indicated by the new evidence than in the consistent condition ($P < 0.0001$). This is hard to explain

if participants are only thinking of the new evidence, and provides substantial additional support to the metacognitive interpretation of the behavior. We have added the following remarks to the general discussion (MM, 7, line 18ff):

“One challenge to our interpretation of this behavior is to suppose that participants are perhaps considering just one piece of evidence at a time, ‘sequentially’, rather than reconsidering their prior belief in light of the new information (27). On this view, participants would only be thinking of the new evidence when they look for more information. To address this concern, we looked at which box participants peeked in first, when they checked for more information. We supposed that if participants are only considering the most recent piece of evidence, then they should be just as likely to peek in the location indicated by the new evidence in both conditions – since the new evidence is identical in both conditions. However, we found that in the ‘Conflicting’ trials participants were significantly more likely to peek in the opposite location to that indicated by the new evidence than in the ‘consistent’ trials ($\chi = 7.93$; $P < 0.0001$) (Figure S5). This is hard to explain if the participants are only thinking of the most recent evidence, and adds weight to our interpretation that the participants are indeed thinking of their prior belief and the new evidence at once.”

And we have added this discussion to the Supplementary Materials, along with the following plot of results (SM, 8, line 20ff)

Figure S5.

Likelihood to check alternative location to new evidence first: Here we see the rate at which participants check the opposite location to that indicated by the new evidence. Boxes represent mean and standard error, bubble size indicates the number of trials on which the alternative location was checked first. It can be seen that in the ‘conflicting’

condition, participants are far more likely to check the opposite box to that indicated by the new evidence than they are in the ‘consistent’ condition. This is hard to explain if we suppose that participants are only thinking of the ‘new’ evidence, since the new evidence is identical in both conditions.

2. Study 1 analysis: The method of using a GLMM works well for one of the conclusions drawn from the analyses (that collapsed across all groups there was an effect of condition overall effect of condition across all groups). However, it is not fit for some of the conclusions made (e.g. which group is driving the overall effect of condition). The finding that there was no interaction between population and condition exemplifies this issue. To then go on and say that the effect is driven by any subset of these groups (e.g. apes/5yos) is unfounded given the statistics presented.

a. There are two reasonable ways to approach this: The first is to use contrast coding within the larger model to look at specific differences between the groups, the other is to run three separate, simpler models on each of the groups.

Thank you. Since there was no interaction in the within-study analyses, it is fair to say that we must be cautious about drawing these conclusions (although the apes clearly fail to significantly distinguish the conditions in the second study, and the three year olds actually peek more in the consistent condition in the individual study – so they cannot be driving the main effect). However, our prediction was that apes would be better in the physical than social task, while children would be better in the social than physical task. This is directly confirmed by the tests for an interaction between condition and study reported below (also see response to reviewer 1) (MM, 6, line 15ff):

“We had predicted that great apes would be more sensitive to a conflict in physical rather than social information, while young children would be more sensitive to conflicting social information. To test this directly we ran a model comparing the conditions across both studies. First we pooled all data and tested for a three-way interaction of population, condition, and study – to see if different groups were more sensitive to the distinction between conditions from one study to the other. This was significant ($\chi^2=11.408$, $df=2$, $P=0.003$). To isolate the source of the three-way interaction we tested for an interaction between study and condition in each population separately. We found an interaction between condition and study in the apes ($\chi^2=4.312$, $df=1$, $P=0.037$). This shows us that the apes distinguished the conditions more in the physical study, than in the social study, or, that apes are more sensitive to contradictory physical information than contradictory social information. We also found an interaction between study and condition in the three-year olds ($\chi^2=11.596$, $df=1$, $P<0.001$). This time the effect is reversed: the three year old children distinguished the conditions more in the social study than in the physical study. There was no interaction in the five-year olds, but rather a main effect of condition: the five-year olds peeked significantly more when faced with conflicting evidence in both studies ($\chi^2=13.746$, $df=1$, $P<0.001$) (see Table T1 for an overview). These results confirmed our hypothesis: apes are more sensitive to conflicting physical evidence than social evidence, while the youngest children in our sample are more sensitive to conflicting social than physical evidence.”

b. In the full model was “condition” significant before removing all of the included variables?

The full-null comparison, where the full model included the terms condition, population, and their interaction, was significant, yes.

In a model-reduction strategy, the procedure is not to read significance levels from the GLMM while all terms are included. Rather, you begin by comparing a full model to a null model that lacks all the terms you are interested in. In our case, the full-null model comparison was significant. This rules out the null-hypothesis. In our full model we included population, condition, and their interaction, and our null-model lacked these terms. Once we know that the full-null comparison is significant, we know that some element of population, condition, or their interaction is significant. On the model reduction strategy, you then go on to identify the source of the significance by checking each component sequentially. The first component to check is the interaction (the highest level term). If there is no significant difference between a model that includes the interaction and one that lacks it, then this shows us that the source of the significance of the full-null comparison is not the interaction. We can therefore discard the interaction, knowing that it is not responsible for the significant difference between the full and null models. The next step is to check whether ‘condition’ is the source of the significance (since we are not interested in, and had no hypothesis about, a main effect of population). And so, we then compare models with and without the term ‘condition’ – looking for a ‘main effect’ of condition. In our case, this was significant. And this is the effect that we have reported (See the following for discussion of this approach: Forstmeier W & Schielzeth H. 2011. Cryptic multiple hypotheses testing in linear models: overestimated effect sizes and the winner’s curse. *Behav. Ecol. Sociobiol.*, 65, 47–55.)

c. Given that the data is clearly not normally distributed, graphing the means for each group/condition does not seem like the most informative graph. A violin plot would highlight the wide variety of responses that were given. OR the type of plots used in figure S1/S2.

The plots used in S1/S2 are actually the same as those in the main manuscript, but lacking boxes. We do not think *removing* the boxes would make the plot more informative. Note that the distribution of responses is represented by the width of the stack of discs at any point on the y-axis – which is exactly what a violin plot would represent by the width of the violin, so it contains the same information. We make this clearer in the caption of the plot:

“Apes and five-year old children sought additional information more when faced with conflicting than consistent evidence, but three-year olds did not. Discs represent individual averages across trials, the number of discs any point on the y-axis represents the distribution of responses. Boxes represent means and standard errors.”

i. Furthermore, on this point a statement about how many animals and children are driving these effects seems necessary. It is clearly not the case that all of the animals look when they get conflicting information (at least 9/18 monkeys, 15/66 5yos, & 18/64 3yos never looked)

1. How many children/monkeys looked more often in the conflicting case?

2. How many actually looked more in the consistent case?

Thank you, these questions require within-subject data which is in study 1 for apes but not children, and study 2 for all groups. The following table gives this information – how many individuals peeked more in one condition rather than the other, or peeked

equally in both conditions; while in study 1 for children we have included the mean peeking for each condition (now added as a supplementary table to SM, 14):

Table S2:

	Apes	3yrs	5yrs
Study 1			
Conflicting	7	0.25	0.3
Consistent	0	0.32	0.18
No Preference	11	n/a	n/a
Study 2			
Conflicting	5	13	15
Consistent	7	3	1
No Preference	4	58	30

Peeking in different conditions: In Table S2, we can see which condition had most peeking in each study and each species. In study 1, 7 apes peeked more in the conflicting than in the consistent condition, while no apes peeked more in the consistent condition. Study 1 was run between subjects for children, and so rather than showing how many children peeked more in one condition than the other we have the means of peeking in each condition. The three year olds actually peek less In study 2, 5 apes peeked more in the conflicting condition, but almost the same number peeked more in the consistent condition. More than 4 times as many 3 years olds peeked in the conflicting condition than in the consistent condition, while fifteen times as many 5 year olds peeked more in the conflicting than in the consistent condition.

d. Given the above questions about the data, the conclusions that “Great apes can detect their own uncertainty... when the reasons for their decision are in conflict.” and “This ability has emerged by age five in human children”, seems like a stretch given that at least 15 5yos never looked and half of the apes never looked. Reporting how many subjects in each group looked more often (compared to how many looked less often) when there was conflicting info is necessary for these conclusions.

We have added the absolute numbers as above. But we think that the high significance of the main effect in study 1 ($P < 0.0001$), which from the plots and numbers is obviously driven by the apes and 5 year olds (since the three year olds actually peeked more in the consistent condition), is enough to support the inferential generalization. But we now have reported the absolute numbers as discussed.

3. Study 2 analysis: My comments about the study 1 analysis all apply to this analysis as well. The only conclusion that can be made given the statistics used is that when all subjects (regardless of group) are collapsed, there is an effect of condition

Our final analysis shows a clear interaction between study and condition, showing that the 3 year olds distinguish the conditions in the social study but not in the individual study – we discuss this result more below (in response to your point 4,a,i).

a. Here the conclusion that 3yos are now successful is unfounded given the statistics used (as is the finding that apes failed).

Please see our discussion of the interaction in our final analysis, below, which justifies this claim.

b. All of the groups seem to have a bias to not look in experiment 1 (regardless of condition), and a bias to look in experiment 2.

i. What is the reason for this?

We think the increased peeking in study 2 is due to increased engagement given the presence of a partner. We have added a point on this to the manuscript (MM 6, line 15):

“It is clear that there is also in general a higher rate of peeking in the second study than the first, which we think is caused by the increased engagement in the task given the presence of a social partner.”

ii. Why is the effect size between the difference tasks (social/not social) much larger than any of the effect seen in condition (consistent/not consistent)?

Thank you. We did not calculate or report effect sizes. If you are referring to the significance levels of the effects, it is true in some cases that the effects are more significant in the tests between-studies than in the tests done within-studies. For example, the main effect of condition in the analysis of 5 year olds between studies is more significant than the main effect of condition within study 1 or within study 2. We think the reason for this is straightforward enough: the 5 year olds clearly distinguish the conditions in both studies, whereas when we look for the main effect of condition within each study, either the 3 year olds (in study 1) or the apes (in study 2), fail to distinguish the conditions, thereby weakening the significance of the main effect.

4. General Discussion

a. The authors address the differences between the specific groups, but do so in a roundabout way that does not completely fix the issues raised in the analysis sections.

i. The statistic used shows there was an interaction effect between experiment & condition in apes only shows that they peeked more in the physical condition compared to the social condition. A better way to do this would be to compare each of these to chance, i.e. is there a significant effect of condition in the physical experiment for apes? Is there a difference between conditions in the social experiment?

Thank you but this is not a correct interpretation of the interaction. If the apes merely peeked more overall from one study to the other, as you have suggested, rather than differently distinguishing the conditions, then there would be *no interaction* between study and condition (there would be a main effect of *study*, but not an interaction between study and condition). The interaction shows that the apes distinguish the conditions more in the individual study than in the social study. In the case of the three year olds, they did the reverse – distinguishing the conditions more in the social than in the individual study.

It is not possible to have a more direct test for this claim. Using multiple t-tests would not test this claim (knowing that one t-test was significant and the other was not

would not tell you if there was an interaction – testing for the interaction directly is the only way to do this). Therefore, the test for the interaction is in fact the most direct test.

1. The above also applies for the conclusion about 3yos.

Reply as above.

ii. Given that these findings are the base of the entire study, more direct analyses are needed.

As explored above, these are indeed the most direct tests for the claims we have made.

b. The explanation for why 3-year-olds are better in the social condition is unclear. It is not the case that experiment 2 was a 100% social task. It was a combination of a social and a physical task. Subjects did not just need to pay attention to social cues, but to the direct physical evidence as well (in which they failed to do in experiment 1). How/why do children pass on the version that integrates social and physical evidence when they fail on physical evidence alone. The explanation that they are better in social-contexts seems incomplete.

Thank you, but it is not correct that in study 2 “Subjects did not just need to pay attention to social cues, but to the direct physical evidence as well (in which they failed to do in experiment 1)”. The contradiction between two pieces of physical evidence does not appear in study 2.

The two studies can be contrasted as follows:

Study 1: participants are given a piece of physical evidence, followed by a subsequent piece of new *physical* evidence (consistent or conflicting);

Study 2: participants are given a piece of physical evidence, followed by a subsequent piece of new *social* evidence (consistent or conflicting).

As you can see in study 2 the participants are not presented with the physical evidence of study 1, so there is no burden to explain why they pay attention to the evidence of study 1 in study 2. They don't.

What 3-year olds pay more attention to is the new piece of social evidence in study 2, which has replaced the new piece of physical evidence from study 1. Our explanation for why the children pay more attention to the social evidence is that human cognition is adapted more to respond to social information than physical information. We solve problems best in pairs or groups, something that is evident from any number of other studies (24-26). And this is exactly how we characterize the expected finding (MM, 2, line 33):

“We expected that since human problem-solving is primarily conducted with social partners (19-20), this ‘social contradiction’ should provide just as strong, or even stronger a cue for children to look for more information than the contradictory physical evidence of the first study. We expected this task would be less compelling for apes, however, whose cognition has evolved primarily for individual problem-solving (21). Finding such a

contrast, we expected, would illustrate the primarily social nature of human cognition, as opposed to the primarily individual nature of ape cognition.”

Minor comments:

- Paragraph 2/line 1: There has also been work showing that monkeys can tell when they have some, but not enough information (see Brady & Hampton, 2021), this is beyond know they have no information.

Thank you, you are right – we have adjusted ‘no information’ to ‘not enough or ambiguous information’ throughout (see also our replies to reviewer 1).

- Were there any differences between ape species?

We did not hypothesize any and we consider it unwise to test for a difference post-hoc. Looking at the plots, the orang-utans seem to do slightly better than the other species in the second study. However with such a small sample size (3 orangutans), and given that orang-utans are less communal than other ape species (suggesting they shouldn’t outperform in a social task) we hesitate to draw any conclusions from this.

- SM/methods & materials: Were the subjects that were excluded counted in the subject count in SM:Participants and the main text? I assume not, but a clarification would be good.

Thank you they were not – we clarify this here by adding a line to the paragraph on exclusion criteria (SM, 2, line 25):

Excluded participants were not included in the participant count reported above.

- What box did the subjects choose when they did not look in the inconsistent trials? Did they choose at random? Or did they choose the most recent information, e.g. the box that showed the item to be larger after the turn?

Thank you that is an interesting question. In inconsistent trials, when subjects do not look:

In the ‘individual study’ their choices matched the more recent piece of evidence in 194 out of 427 trials (45%). This indicates no particular preference for the older or newer evidence.

In the ‘social study’ in 112 out of 161 trials (69%), they chose the box that the first piece of evidence indicated was the correct choice. This suggests that the first (perceptual) evidence was more influential.

This is not surprising given that in the first study, the second piece of evidence is in the same ‘modality’ as the first (direct perceptual evidence), while in the second study, the second piece of evidence is the preference of the social partner. We have added these descriptive statistics to the supplementary materials (SM, 8, line 40).

- Table S1. Seems uninformative as is. Instead of broad conclusions, including the direct statistics in these boxes would be much better. E.g. each of the boxes represent the results of a model using “looking” as the dependent variable and condition as the predictor. That way

you have 6 models each summarized in a box rather than a conclusion without any stats to back it up.

Thank you. Note that we do not have six but rather five pertinent analyses – two exploring populations across each study; and one for each population looking at interactions between the studies. So, it does not make sense to put a p-value in each cell. We have however added boxes to the table ‘between’ cells, to capture the interaction between condition and study for each population, which are the results that speak most directly to our hypotheses:

Table S1.

More information-seeking given:	Apes	Children, 3yrs	Children, 5yrs
Conflicting Physical Evidence	Yes	No	Yes
	P = 0.037	P < 0.001	P < 0.001
Conflicting Social Evidence	No	Yes	Yes

References:

Brady, R. J., & Hampton, R. R. (2021). Rhesus monkeys (*Macaca mulatta*) monitor evolving decisions to control adaptive information seeking. *Animal Cognition*, 1-9.

-Stephen Ferrigno

Sincere thanks for the effort you have put into our paper, it has been greatly improved by your proposals.

.....

Referee: 3

Comments to the Author(s)

The paper presents some clever manipulations that reveal some interesting aspects of human and nonhuman information-seeking behavior. But, it is seriously marred by an unnerving and unjustified predilection toward over-interpretation and using mentalistic notions when simpler behavioral interpretations would suffice. Many terms and expressions are set in quotation marks leaving the reader to wonder if these are scientifically suitable notions or just window dressing to attract greater public attention.

Thank you. It is always a tricky question how much interpretation to give of observational data. We have aimed at achieving an ‘Ockham’ like balance of postulating the most plausible internal mechanism that explains the observational data, without positing internal mechanisms that are unnecessary.

Every mentalistic term we have used is well-established in the literature, given the kind of observations we have made, with the one exception of the expression ‘rational monitoring’. We have coined this expression as an adaptation of ‘metacognitive monitoring’ in order to capture the behavior described, which goes beyond what previous studies on metacognitive monitoring have shown. This use of ‘rational’ is consistent with theories that treat rationality as the monitoring of grounds for making decisions (see for example List and Pettit 2011, chp1, cited in main manuscript). We think the most plausible internal process to explain the newly documented behavior is internal monitoring of the prior belief against the new evidence – evaluating two conflicting reasons for the decision – hence the expression.

The paper would definitely profit by the authors taking a more circumspect approach and more seriously considering the manipulations and findings from a behavioral perspective. After all, we only have what can be observed and confirmed--the rest is sheer speculation.

Thank you for the proposal. We are happy to explore other competing explanations and have revised our general discussion taking this advice into consideration, along with challenges raised by reviewers 1 and 2 (MM, 7, line 18ff):

“One challenge to our interpretation of this behavior is to suppose that participants are perhaps considering just one piece of evidence at a time, ‘sequentially’, rather than reconsidering their prior belief in light of the new information (27). On this view, participants would only be thinking of the new evidence when they look for more information. To address this concern, we looked at which box participants peeked in first, when they checked for more information. We supposed that if participants are only considering the most recent piece of evidence, then they should be just as likely to peek in the location indicated by the new evidence in both conditions – since the new evidence is identical in both conditions. However, we found that in the ‘Conflicting’ trials participants were significantly more likely to peek in the opposite location to that indicated by the new evidence than in the ‘consistent’ trials ($\chi = 7.93$; $P < 0.0001$) (Figure S5). This is hard to explain if the participants are only thinking of the most recent evidence, and adds weight

to our interpretation that the participants are indeed thinking of their prior belief and the new evidence at once.

Another concern that could be raised is that perhaps, when faced with conflicting information, an involuntary hesitation might become a cue that participants could use to learn that whenever they experience it, they will do better to look for more information before choosing (13). On this view participants would simply react to a ‘feeling of uncertainty’ that they use as a cue to information-seeking. Alternatively, we might worry that this behavior could be generated by a decision taking too long, triggering information-seeking as a sort of ‘reset’ to return to foraging (28). On both of these interpretations, participants do not know what they do not know, but are instead engaged in involuntary information-seeking triggered by a cue. Arguably if this were true we should expect that they would search randomly for food when their information-seeking began – instead of looking for exactly the information they need to answer the question they are uncertain about. The participants in the current studies, however, do not search at random; they look for exactly the information they need to resolve the conflict (peeking inside the containers they have already been exploring). This ‘targeted information-seeking’ (3) is best explained by supposing that participants recognize what it is they do not know, and are seeking the information they need to specifically address this question.”

Furthermore, the paper is not well written. There are a few too many grammatical and punctuation errors as well as a serious mislabeling in the caption of Fig. S1. More careful preparation of a revision would certainly be in order.

Thank you for highlighting that error, it is corrected, as we believe are any other typographical errors.

Finally, there is something of an unholy blending of within-subject and between-subject manipulations finding their way into the same statistical analyses.

Within and between-subject data are routinely combined in mixed models, which are designed to handle this: the models control for individual ID, therefore preventing the inflation of power in the between-subject data modelled alongside the within-subject data (See Seltman H. (2018) *Experimental Design and Analysis*, Carnegie Mellon University Press, chapter 15, for discussion).

There is also concern that the two studies might not properly be included in a single statistical analysis.

Here the question is conceptual: are the studies similar enough, apart from the variable measured in the models that compare performance between studies (i.e. whether the new evidence is physical or social)? We think that they are:

Study 1: participants are given a piece of physical evidence, followed by a subsequent piece of new *physical* evidence (consistent or conflicting);

Study 2: participants are given a piece of physical evidence, followed by a subsequent piece of new *social* evidence (consistent or conflicting).

We believe this straightforward parallel between the studies warrants their direct comparison.

And, there are very few individuals from some ape species in the studies and their data are also quite disparate; can they all legitimately be lumped into a single 'ape' group?

Our main hypothesis concerned the difference between young children and apes. We did not on the other hand have any hypotheses about differences between ape species, so any separation into distinct groups for the sake of analysis would be ad hoc, we think, and also, as you have pointed out, difficult to defend given the small numbers for each species.

Thank you very much for your time in reading our paper, we believe it has been substantially improved by your critique.

Appendix B

Replies to Associate Editor and Reviewers

Note: Text in blue is the reviewer's commentary. Indented in black is our reply, with excerpts from the paper in red.

Reply to Associate Editor: Page 2

Reply to reviewer 1: Page 4

Reply to reviewer 2: Page 9

Associate Editor

Comments to Author:

This is a revision of a paper examine the evolutionary and developmental origins of metacognitive reasoning, implementing a novel task to test what kinds of information children and apes use to assess what they know. The reviewers and I agree that this is a responsive revision that has greatly improved the paper. However, the reviewers also have several helpful comments on new version of the manuscript that should still be clarified. As an additional point, I agree that the new coding and analysis of location of searching is a nice addition, but I think this primary reporting of the coding and analyses of this measure needs to be moved to the methods and results section rather than reported for the first time in the general discussion.

We thank the editor for their careful handling of our paper and review process. We are delighted the editor is happy with the new analysis, and we have moved it as suggested into the results section (MM, 6):

“One challenge raised for studies like this one is that participants may be considering just one piece of evidence at a time (Leahy & Carey 2020). This would mean participants were thinking only of the new evidence when they sought more information, and that would not be metacognition. To investigate this we checked where participants looked first when peeked. If participants only considered the new evidence, then they should be just as likely to first check the location indicated by the new evidence in both conditions – since the new evidence is identical in both conditions. On the other hand, if participants are considering their prior belief and the new evidence at once, then when the two conflict (Conflicting) they should be more likely to first check in the opposite location to that indicated by the new evidence, than when the two indicate the same location (Consistent). We found that in the ‘Conflicting’ trials participants were indeed more likely to peek first in the opposite location to that indicated by the new evidence than in the ‘Consistent’ trials ($\chi = 7.93$; $P < 0.0001$) (**Fig.S3**). This is hard to explain if participants were not thinking of their prior belief and the new evidence at once.”

A new concern raised by reviewer 1 is that the apes might not be paying attention to their partners in the social study. In fact we had conducted an ‘ignorance-knowledge’ post-test to check exactly this in the original study, although we had not discussed it in the main manuscript. When apes do not know anything about the location of a reward (as opposed to having a prior belief), they follow their knowledgeable partner’s choice (as do 3 and 5 year olds). This excludes the possibility that they are not paying attention to their partner or do not think their partner is helpful. We have now discussed this test in the main manuscript to highlight it (MM, 6):

“Another concern is that in the social study, the apes may not be attending to their partner’s choices. This would explain why they were slower to recheck evidence in light of conflicting opinions than conflicting physical evidence. To rule this out we conducted an ‘ignorance-knowledge’ post-test. Here participants received no prior information about the location of the reward, but could see that their partner could see where it was. Now participants significantly followed their partners’ choices (5yr mean = 0.91, $P < 0.0001$; 3yr mean = 0.88, $P < 0.0001$; Apes mean = 0.63, $P < 0.001$) (**Fig. S4**). This rules out the possibility that the apes were not paying attention to their partners in the social study. Apes knew what choices their partners were making, but peer disagreement was not enough to get them to doubt their prior belief.”

The second reviewer thinks the paper should be published, but that we should remove some of the main claims. We disagree, since we believe they are soundly justified by the tests we have used. They have alternatively suggested that we might remove the table of main findings from the supplementary materials; we can do so, but we think it is a helpful table – we leave this to the discretion of the editor.

We have addressed the reviewers' other minor comments below. We hope the manuscript is now ready for publication.

Reviewer(s)' Comments to Author:

Referee: 1

Comments to the Author(s).

I thank the reviewers for addressing my comments and providing new analysis and a careful examination of potential alternative interpretations, the manuscript has greatly improved in my opinion.

The new analysis reporting where participants look after seeing conflicting new evidence is great! I believe this is now the first study that is able to fully address the 2 concerns I raised during the first round of review in both children and apes simultaneously... This has been a pervasive problem in developmental and comparative studies of metacognition for quite some time, so perhaps this aspect could be emphasized more in the discussion by dedicating a paragraph to this issue, and referring to past findings in both children & apes?

We are delighted the reviewer likes this analysis – and we thank them for the challenges that lead us to conduct it. As recommended we have moved the discussion of this analysis to the main section and expanded the paragraph in which it is discussed:

“One challenge for studies in which several pieces of evidence are presented in succession is to suppose that participants may be considering just one piece of evidence at a time (Leahy & Carey 2020). In the current studies this would mean participants were thinking only of the new evidence when it is presented, and not comparing it to their prior belief – and this would not be metacognition. To address this, we analyzed which box participants peeked in first, whenever they peeked. We supposed that if participants were only considering the newer evidence when they saw it, then they should be just as likely to peek in the location indicated by the new evidence in both conditions – since the new evidence is identical in both conditions. But if they were considering their prior belief along with the new evidence, then when the two conflict they should be more likely to first check in the alternative location to that indicated by the new evidence. We found that in the ‘Conflicting’ trials participants were indeed significantly more likely to peek first in the opposite location to that indicated by the new evidence than in the ‘Consistent’ trials ($\chi = 7.93$; $P < 0.0001$) (**Fig.S4**). This adds weight to our interpretation that the participants are indeed thinking of their prior belief and the new evidence at once.”

I still have one comment about some of the interpretation of the findings. This is perhaps only related to wording and should be easy to fix:

“apes were more sensitive to conflicting physical evidence than social evidence”

This is a strong conclusion, and I am still wondering about one slightly alternative interpretation: is it possible that the cross-species difference is due to more general differences in the way individuals see other individuals by default (e.g., as cooperative or potentially deceitful)? Could it be that children see one another as genuine cooperative “partners” by default, while apes see other agents as potential competitors? “In this case withholding judgment and looking for more information requires the ability to understand not only that our own beliefs and the information it was based on might be false, but also that our

partner's belief may be false": AND it requires evaluating the intentions of the other agent (i.e., is the partner likely to disclose information truthfully or not)? Would this interpretation in terms of social status/relationships not be more parsimonious, as it does not imply that apes would never consider conflicting social information, but rather, that they are not necessarily going to believe what a social "partner"'s behavior discloses about the location of a reward? Would this not be compatible with what we believe about apes' social cognition and some of the last authors' claims about the unique pro-sociality of humans? I can see how it may be argued that this is actually less parsimonious as it assumes that apes have the ability to attribute deceitful intents to the other agent? In any case, it would be good to discuss this.

Thank you for this point. We had the same concern, and were worried that the apes might in general ignore their partner's choices. This would undermine our claim that they cannot incorporate their partner's conflicting opinion into their own decisions, since it may be the case that they simply are not paying attention. To check this, we presented apes and children with an 'ignorance-knowledge' post-test (something we included in the original submission, but might not have emphasized clearly enough). Here participants are ignorant of the location of the reward entirely (they have no evidence) but they can see that their partner knows where the reward is. We found that both apes and children significantly followed their partner's choices in these cases. This shows that it cannot be the case that the apes are completely ignoring their partners' choices, or that they consider them entirely unhelpful (since they copy their partner's choice when they have no competing belief themselves). Instead, it only remains to conclude that they know what their partner has chosen, but when their partner's choice conflicts with their own they do not take this as a reason to doubt their own view.

We have now discussed this test in the main text (MM7):

““Another concern is that in the social study, the apes may not be attending to their partner's choices. This would explain why they were slower to recheck evidence in light of conflicting opinions than conflicting physical evidence. To rule this out we conducted an 'ignorance-knowledge' post-test. Here participants received no prior information about the location of the reward, but could see that their partner could see where it was. Now participants significantly followed their partners' choices (5yr mean = 0.91, $P < 0.0001$; 3yr mean = 0.88, $P < 0.0001$; Apes mean = 0.63, $P < 0.001$) (**Fig. S4**). This rules out the possibility that the apes were not paying attention to their partners in the social study. Apes knew what choices their partners were making, but peer disagreement was not enough to get them to doubt their prior belief.”

Minor

p2.: I am still not super convinced that the current study targets a different competence as compared to past studies in the field of metacognition, i.e., that weighing uncertainty during decision making vs. revising beliefs in light of new conflicting evidence must involve strictly different processes (e.g., see work showing a continuity between the mechanisms supporting changes of mind in light of dynamically evolving evidence and confidence judgments, e.g., by Kiani, Shadlen and colleagues, or Fleming and colleagues etc...). One way to clarify this would be to refer to the distinction between prospectively evaluating competing options and confidence before making a choice, and retrospectively using past-choice/evidence to make subsequent decisions (i.e., prospective vs. retrospective metacognition). It is also the case that

if you have to make a choice between an option A or B – for instance in the tone example - the tone can only be A or B, so there is a conflict / an incompatibility there too...

Thank you for this worthwhile challenge, let us try to elaborate further.

The A or B studies you mention involve deciding between two conflicting options. But our study involves making a decision that conflicts with a *prior decision*. When presented with two options, and unable to choose both (e.g. A or B but not both), it is true that one option ‘conflicts with’ the other. But at no point in such a scenario does the participant have to make a choice/ form a belief that conflicts with their own prior choice/belief. There is no conflicting choice/belief because there was no prior choice/belief. And so a task involving conflicting options does mean participants have to think about a prior choice/belief.

Also, as you say, some studies explore updating beliefs in light of new evidence (e.g. Legare et al, cited). If I believe it is sunny and then look out the window and see that it is raining, I might acquire a new belief (“it’s raining now”). But this need not involve metacognition – I can acquire a new belief without thinking about what I believed before (I need not go on to reflect “but before I believed it was sunny!”). The double-checking in our study, on our view, is triggered because the participants know what they believed before, and recognize the conflict between their prior belief and the new evidence, prompting them to doubt the new evidence and double-check. The new analysis (that your previous challenge prompted us to undertake!) further supports this interpretation – participants are indeed thinking of their prior belief, since they check first in the location indicated by their prior belief. Studies illustrating the acquisition of new beliefs in light of new evidence, on the other hand, do not show that participants are thinking about their prior beliefs, and how they relate to the new ones. The mechanisms involved may be the same (if every time we acquired a new belief, this involved metacognition), but we doubt it (or, every animal that can acquire new beliefs would be engaged in metacognition).

We have tried to capture this distinctiveness by further revising the introduction:

“While these studies reveal the ability to detect when one has no knowledge, they do not reveal an ability to think about what one already believes, which is a different kind of metacognition. **Thinking about what one believes is sometimes elicited when one encounters new evidence that calls a prior belief into question. Suppose one believes it is sunny out, and plans to go to the beach. Now the dog comes into the house soaking wet. This may prompt us to look out the window to recheck our grounds for believing it is sunny before we decide to leave. In such a scenario we are aware of what we believe, and we check the grounds or reason for that belief against what the new evidence tells us.** Following terminology used by philosophers (14) this can be called ‘rational’ or ‘reason-based’ monitoring of the decision-making process.”

And the general discussion:

“Previous studies have shown that several species will look for more information when they do not have enough to answer a question (1-11), and that young children will update their beliefs to match new evidence (22-26). But our studies show for the first time that

apes and young children seek more information when old and new evidence conflict, but are equally strong. **Rather than simply updating their earlier belief to match the new evidence, they double-check the evidence – checking first in the location indicated by their prior belief. The intuitive explanation for this behavior is that participants knew what they believed, and sought to compare the reason for their prior belief with what the new evidence told them, recognizing that either could be wrong.** They were, in other words, examining the reasons for their belief or ‘rationally monitoring’ the decision-making process. Apes were more sensitive to conflicting physical evidence rather than peer disagreement, while young children were more sensitive to peer disagreement.”

p.6: new description of the 3-way interaction, thanks this is really useful. Could you also provide the stats for the main effect of condition here for both the apes and the 3yo separately for each study?

We are very happy the three-way interaction analysis is now clearer.

We understand the temptation to retest further subsets of the data, but we do not think it is a good idea. The main effect of condition in each study justifies the conclusion that overall there is an effect; the interaction between studies justifies that sensitivity to the type of contradiction varies between populations. For example, the interaction between study and condition in the apes shows that the apes must distinguish the conditions in one of the studies more than the other; but it is not possible for them to distinguish the conditions ‘more’ in the individual than the social study without distinguishing them in the individual study. Therefore the interaction already supports the conclusion that the apes distinguish the conditions in the individual study. With those results in hand, the best practice, we believe, is to decide on the best interpretation of these results by inspecting the plots. On the other hand, repeatedly testing the data for the same hypothesis undermines the coherence of p-value testing, and further reducing the data set could eliminate the effect for no reason other than having reduced the power, resulting in a misleading outcome.

Thanks for revising the description of the statistics. This is clearer, but still not 100% clear, perhaps just include the formulas in the methods and/or SM for readers to check this if they are interested? and or simply mention that the formulas are available on OSF somewhere in the description of the stats.

Thank you, the full models/formulas were indeed already included on the OSF page, as stated in our Data availability statement:

“Data and materials availability: Data, statistical models, analyses and sample photographs of material used are available online at the Open Science Foundation website: <https://osf.io/ey5f9/>.”

We have also added a reference to the OSF page early in the main text in section 1 (MM3):

“We used Generalized Linear Mixed Models (GLMMs) with binomial error structure to examine whether participants distinguished the conditions (SM, 6-9; models and data available at <https://osf.io/ey5f9/>).”

“yes, children are better at monitoring their own knowledge and recognize that they lack information in the social study – just as you have suggested »

Yes, but a lot of the directly relevant literature on the development of metacognition is still not covered in the paper to reflect the fact that we already thought this was the case; e.g., studies by Kim, Sodian, Proust and colleagues have already shown that young children's metacognition is potentiated in social contexts...

A space restriction prevents us from including many more references. But we have added Kim et al. now in addition to the Shea already added.

The discussion is really short and - also see above - it would be nice to expand a bit on some aspects. e.g., "This fits with theoretical models that take human metacognition to be adapted for solving peer disputes (31), (but the claim is somewhat wider than this...? i.e., metacognition has a social function, not simply for solving conflicts, but also other purposes like knowledge transmission etc., for instance see also papers by Heyes, or Dunstone and Caldwell...) and that human rationality is originally social (19-21)."

I don't know if the findings fit so comfortably with this, as they also show that reasoning is not typically human...? So how does that relate to claims (e.g., by Sperber/Mercier or others that are already cited in the paper) that reasoning evolved to fulfill a social function? It could be good to expand a little bit on this.

That's an excellent point. Our view is that *human* rationality has a social component, not that rationality evolved exclusively for the social purposes to which humans put it. In light of the space constraints it is difficult to get into much more detail but we have altered the sentence you noted to capture this:

"This fits with theoretical models that take human rationality to be adapted for social purposes, including solving peer disputes, argumentation, and knowledge transmission (19-21, 31, 32)."

"but are instead engaged in involuntary information-seeking triggered by a cue" do you mean automatic rather than involuntary?

Yes, thank you – we've changed this.

We would like to take the opportunity to once again thank the reviewer for such extensive and helpful comments on our work.

Referee: 2

Comments to the Author(s).

The authors have done a great job addressing my concerns and I think the manuscript should be published.

Thank you again for your careful review and your support. We are happy we have addressed your concerns and that you think the paper is now ready to be published. We discuss the comment about statistics below.

However, I do still have one minor comment about the statistics used to draw conclusions about success and failure within each study for a given population.

My concern is exemplified in Table S1. The statistics used here (the interaction between condition & study for each population) is the correct statistic to show that there was a difference between conditions and rightly justifies the conclusions that apes are more sensitive to physical than social cues, and 3-year-olds are more sensitive to social than physical cues. However, there is no direct tests of success or failure on the individual studies (e.g. the “Yes” and “No” that is included in the table). The only statistics used within a study show that collapsed across all groups, there is a significant effect of condition in both Exp. 1 & Exp. 2. This does not speak to success or failure of individual populations within a group. The two passages below make conclusions about results from specific populations within a study that currently lack statistical evidence:

“Great apes can detect their own uncertainty not only when they have no or ambiguous information (8- 11), but when the reasons for their decision are in conflict” pg.4 line 12-14, “The results of our second study indicate that children as young as three take peer-disagreement as a reason to call their beliefs into question.” pg.6 line 8-9

If the authors want to make these claims, I suggest they run pairwise comparisons between conditions for each study (e.g. testing if the effect of condition in Study 1 is significant and likewise with Study 2). This can be done using the same regression already being implemented on page 6 & SI “Studies 1 and 2 compared” for separate populations.

Thank you. We understand the reviewer’s point, however, the claims above are indeed supported by the main effects in each study taken separately, and reinforced by the interactions between studies. That “Great apes can detect their own uncertainty ... when the reasons for their decision are in conflict” is supported both by the main effect of study 1, and the interaction between condition and study as reported. Consider : it is clear from the interaction that the apes distinguish the condition more in the individual study than in the social study; but if they distinguish the conditions ‘more’ in the individual than the social study, then this entails that they must distinguish the conditions in the individual study. So the interaction already establishes this claim. Similarly, that “children as young as three take peer-disagreement as a reason to call their beliefs into question” is again supported both by the main effect of study 2, plus the interaction between studies.

We think the most sound statistical strategy is to take the results of the main effects of study and the interactions between studies, and offer the best interpretation of them based on inspecting the plots, rather than repeatedly test smaller sets of data for each aspect of the interpretation – the latter undermines the coherence of p-value testing,

and risks eliminating effects because of the reduction in power (even though the claims are already established).

Alternatively, the passages noted above (and the “Yes”/“No” in Table S1) could be removed.

The words ‘yes’ and ‘no’ in this table indicates our interpretation of the results of the studies. There is evidence to support the conclusion that apes are more sensitive to the difference between conflicting physical evidence than social evidence; and evidence that 3 year olds are more sensitive to conflicting social evidence than physical evidence. This is what the table indicates. We have adjusted the table legend from reading “Overall we found that great apes...”, to:

“Overall we take the evidence presented to indicate...”

However if the editor thinks the table should be removed we can do that.

I would like to note that a majority of the conclusions drawn are distinctly about the differences between Study 1 & 2 (for the different populations) which has been clearly established by the authors using the interaction of study*condition for each population. These conclusions are well warranted given the data and analyses used.

We are happy the reviewer is satisfied with these analyses. Many thanks again for your patient work on our paper, it is greatly appreciated.